# Structural titration reveals Ca²⁺-dependent conformational landscape of the IP₃ receptor

Navid Paknejad ⬤ [1,2,3], Vinay Sapuru ⬤ [1,2,3] & Richard K. Hite ⬤ [1] ✉

Inositol 1,4,5-trisphosphate receptors (IP₃Rs) are endoplasmic reticulum Ca²⁺ channels whose biphasic dependence on cytosolic Ca²⁺ gives rise to Ca²⁺ oscillations that regulate fertilization, cell division and cell death. Despite the critical roles of IP₃R-mediated Ca²⁺ responses, the structural underpinnings of the biphasic Ca²⁺ dependence that underlies Ca²⁺ oscillations are incompletely understood. Here, we collect cryo-EM images of an IP₃R with Ca²⁺ concentrations spanning five orders of magnitude. Unbiased image analysis reveals that Ca²⁺ binding does not explicitly induce conformational changes but rather biases a complex conformational landscape consisting of resting, pre-activated, activated, and inhibited states. Using particle counts as a proxy for relative conformational free energy, we demonstrate that Ca²⁺ binding at a high-affinity site allows IP₃Rs to activate by escaping a low-energy resting state through an ensemble of preactivated states. At high Ca²⁺ concentrations, IP₃Rs preferentially enter an inhibited state stabilized by a second, low-affinity Ca²⁺ binding site. Together, these studies provide a mechanistic basis for the biphasic Ca²⁺-dependence of IP₃R channel activity.

Inositol-1,4,5-trisphosphate receptors (IP₃Rs) are large, tetrameric cation channels that serve as the primary intracellular calcium (Ca²⁺) release channels in nonexcitable cells. Expressed in the endoplasmic reticulum (ER), IP₃Rs mediate the flow of Ca²⁺ from the ER into the cytoplasm and other cellular compartments where Ca²⁺ contributes to the regulation of cell division[1], differentiation[2], metabolism[3], migration[4,5], and cell death[6]. Consequently, dysregulation of IP₃Rs is associated with numerous pathologies including cancer[7–9], neurological[10,11], cardiac[12], and immune[13] diseases. IP₃R activation requires nanomolar cytosolic Ca²⁺ and the second messenger IP₃, whose production is stimulated by receptor tyrosine kinase and G protein-coupled receptor signaling pathways[14–20]. Notably, IP₃Rs are inhibited by micromolar cytosolic Ca²⁺ concentrations, resulting in a biphasic dependence on Ca²⁺ for channel activity. The recursive nature of IP₃R regulation by its permeant ion results in the emergent phenomenon of Ca²⁺ oscillations in cells. The Ca²⁺ dependence of both activation and inhibition are further modified by the concentration of IP₃ as well as ATP, ER Ca²⁺ and numerous protein interaction partners[21–23]. In this manner, IP₃Rs integrate multiple upstream signals to tune the frequency and amplitude of Ca²⁺ oscillations that encode regulatory information for diverse cellular processes such as mitochondrial oxidative metabolism[24], gene expression[25], lymphocyte activation[26] and neuronal development[27].

Structural snapshots of IP₃Rs have revealed the overall architecture of the channel and how IP₃ and Ca²⁺ can stabilize conformational changes[28–34]. These studies revealed that IP₃Rs possess a transmembrane domain that resembles other 6 transmembrane (6TM) ion channels such as voltage-gated ion channels and TRP channels, as well as a large cytosolic domain (CD) that contains all of the known regulatory ligand-binding sites, including two Ca²⁺ binding sites, and shares some homology with the Ryanodine Receptor (RyR)[35]. When both Ca²⁺ binding sites are occupied, the pore remains closed regardless of IP₃ binding status[29]. In contrast, a recent structure suggests that the pore opens when only one of the Ca²⁺ binding sites is occupied in the presence of IP₃[33]. However, many additional conformations have been resolved whose functional corollaries remain

[1]Structural Biology Program, Memorial Sloan Kettering Cancer Center, New York, NY 10065, USA. [2]Physiology, Biophysics, and Systems Biology (PBSB) Program, Weill Cornell Graduate School of Biomedical Sciences, 1300 York Avenue, New York, NY 10065, USA. [3]These authors contributed equally: Navid Paknejad, Vinay Sapuru. ✉e-mail: hiter@mskcc.org

unclear. More broadly, the conformational landscape that enables IP$_3$Rs to pivot from activation to inhibition to generate Ca$^{2+}$ oscillations remains unknown. Here, we collect electron cryomicroscopic images of human type 3 IP$_3$R (hIP$_3$R3) vitrified in a broad range of Ca$^{2+}$ concentrations and treat particle abundance as a proxy for the relative free energy of each state to establish high-resolution thermodynamic models of IP$_3$R activation and inhibition, which combined with cellular Ca$^{2+}$ imaging elucidates the structural basis for IP$_3$R-generated Ca$^{2+}$ oscillations.

## Results

### Structural Ca$^{2+}$ titration reveals conformational landscape of hIP$_3$R3

To elucidate the mechanisms by which IP$_3$ and Ca$^{2+}$ together activate the channel, and high Ca$^{2+}$ concentrations inhibit the channel, we collected transmission electron cryomicroscopic (cryo-EM) images of purified human type 3 IP$_3$ receptors (hIP$_3$R3) prepared with saturating (200 μM) IP$_3$, saturating (1 mM) ATP, and five concentrations of nominal free Ca$^{2+}$ spanning a range from 1 nM to 10 μM (Fig. 1a and Supplementary Fig. 1). Our cryo-EM conditions correspond to a range where electrophysiological analyses predict that hIP$_3$R3 displays a biphasic relationship between Ca$^{2+}$ concentration and channel open probability. To track the Ca$^{2+}$-dependence of the IP$_3$R conformational landscape in an unbiased manner, we merged these datasets and performed image processing in aggregate (Supplementary Fig. 2 and Supplementary Table 1).

Using hierarchical classification without imposing symmetry, we resolved five major states for hIP$_3$R3 at resolutions up to 2.5 Å to which we initially assigned C4 symmetry (Supplementary Table 2). By relaxing our assumption of C4 symmetry and computing latent representations of the conformational heterogeneity present in the remaining classes using 3D variability analysis (3DVA)[36], we were also able to reconstruct discrete low-abundance intermediates, including several that are asymmetric. Following classification, we improved the interpretability of the reconstructions by performing symmetry expansion and local refinements that were subsequently merged into composite reconstructions.

Due to overlapping ligand-binding profiles of the major states and several minor states, we established a heuristic describing four features of the channel that facilitate comparisons between the states as well as with existing IP$_3$R structures. The features that comprise the heuristic are the beta-trefoil (BTF) ring, armadillo repeat domain 2 (ARM2), the juxtamembrane domain (JD) ring and the pore (Fig. 1b–f). The most predominant of these features is the conformation of the cytosolic BTF ring, which adopts either an intact tetrameric ring structure that stabilizes the entire cytosolic domain (CD), or a disrupted state in which the CDs of the four protomers are decoupled and highly dynamic. Second is the conformation of the peripheral ARM2 domain, which can be either extended away from the rest of the CD or retracted. Third is the JD ring, located at the interface between the CD and the transmembrane domain (TMD) and can adopt either an intact ring structure or a disrupted, open conformation. Last is the pore, which can either be closed or open.

In the first of the major states, the BTF ring is intact, ARM2 is extended, the JD ring is intact, and the pore is closed (Fig. 1b, Supplementary Fig. 3 and Supplementary Table 2). As this state resembles previously published ligand-free states of IP$_3$Rs in various detergents (PDB: 3JAV, 6DQJ, 6MU2, 6UQK, 7LHF) and lipid environments (PDB: 7LHE)[28–32], we assigned this conformation as a resting state. Two similar minor states were also present that share the overall conformation of the resting state but differ slightly in the conformation of the TMD with much weaker density for the peripheral S1-S4 domain (Supplementary Fig. 4 and Supplementary Table 3). Due to the increased conformational heterogeneity of the TMD in these states, we assigned them as labile resting state 1 and labile resting state 2.

The second and third major states also have intact BTF and JD rings and a closed pore, but their ARM2 domains adopt the retracted conformation, where ARM2 is rotated towards the central linker domain (CLD) (Fig. 1c–d, Supplementary Figs. 5 and 6 and Supplementary Table 2). Differentiating these two states is the presence of a non-protein density occupying the previously identified JD Ca$^{2+}$ binding site that we assigned as a bound Ca$^{2+}$ ion. A fourth state shares the intact BTF ring and retracted ARM2 domain with the second and third states, but its pore is open, and its JD ring is disrupted (Fig. 1e, Supplementary Fig. 7 and Supplementary Table 2). Based on the open conformation of the pore, we assigned the fourth state as an activated state. This activated state is similar to a recently published structure of hIP$_3$R3 with its pore in an open configuration (PDB: 7T3T)[33]. As the second and third states differ from the activated state only in their closed pores and intact JD rings, we assigned them as a preactivated state and a preactivated+Ca$^{2+}$ state, respectively. The preactivated state, with its intact BTF ring and retracted ARM2 domain and closed pore, is similar to previously reported structures of IP$_3$Rs in the presence of IP$_3$ (PDB: 6DQV, 7T3P, 7T3Q, 7T3R)[33,37], but the preactivated +Ca$^{2+}$ state has not been previously described.

In addition to the four-fold symmetric resting and preactivated states, we also resolved classes with asymmetric CDs. In these classes, either one, two or three of the ARM2 domains adopt the retracted conformation (see below and Supplementary Table 4). Together, these classes represent a continuum of states between the resting state, where all four ARM2 domains are extended, and the preactivated state, where all four ARM2 domains are retracted, a finding we previously reported for channels in the presence of IP$_3$ (PDB: 6DQN, 6DQS, 6DQZ, 6DR0, 6DQV)[29]. While we were able to resolve structures for these states, we observed significant continuous heterogeneity among these asymmetric classes. Therefore, we combined these particles into an ensemble that we call the resting-to-preactivated transitions for quantification. We also observed classes with asymmetric features in the JD and TMD that otherwise resembled the resting or preactivated states. The pore in these classes has undergone movements that result in either two-fold pseudosymmetric (-C2) or four-fold pseudosymmetric (-C4) dilations compared to the closed states (see below, Supplementary Fig. 8 and Supplementary Table 5). We will refer to the classes with extended ARM2 domains as resting TMD transitions and those with retracted ARM2 domains as preactivated TMD transitions.

In the fifth major state, the BTF ring is disrupted, ARM2 is retracted, the JD ring is intact, and the pore is closed (Fig. 1f, Supplementary Fig. 9 and Supplementary Table 2). A minor population of particles sharing these features was also identified in which the channels were organized into higher-order assemblies containing two or more tetrameric channels (see below, Supplementary Fig. 10 and Supplementary Tables 3, 10, and 11). Notably, the interactions that mediate the assemblies are the only distinguishing feature between these two states. Otherwise, the channels adopt similar conformations. These two BTF ring disrupted states are reminiscent of previously published Ca$^{2+}$-bound hIP$_3$R3 structures (PDB: 6DRC, 6DR2, 6DRA, 7T3U)[29,33], where BTF ring disruption confines IP$_3$-mediated conformational changes to the CD, so we assigned the major state as an isolated inhibited state and the minor state as a higher-order assembly of inhibited states.

### Ligand dependence of hIP$_3$R3 conformations

To evaluate the relationship between ligand occupancy and conformational state, we inspected the cryo-EM maps and identified densities in the resting, preactivated, preactivated+Ca$^{2+}$, activated and inhibited states consistent with an IP$_3$ bound at the BTF2-ARM1 interface and with a Zn$^{2+}$ and an ATP bound in the JD of all five C4 symmetric major states (insets in Fig. 1b–f). Density for IP$_3$ is also present in the asymmetric subclasses that belong to the resting-to-preactivated transitions (Supplementary Fig. 11), indicating that the 200 μM IP$_3$

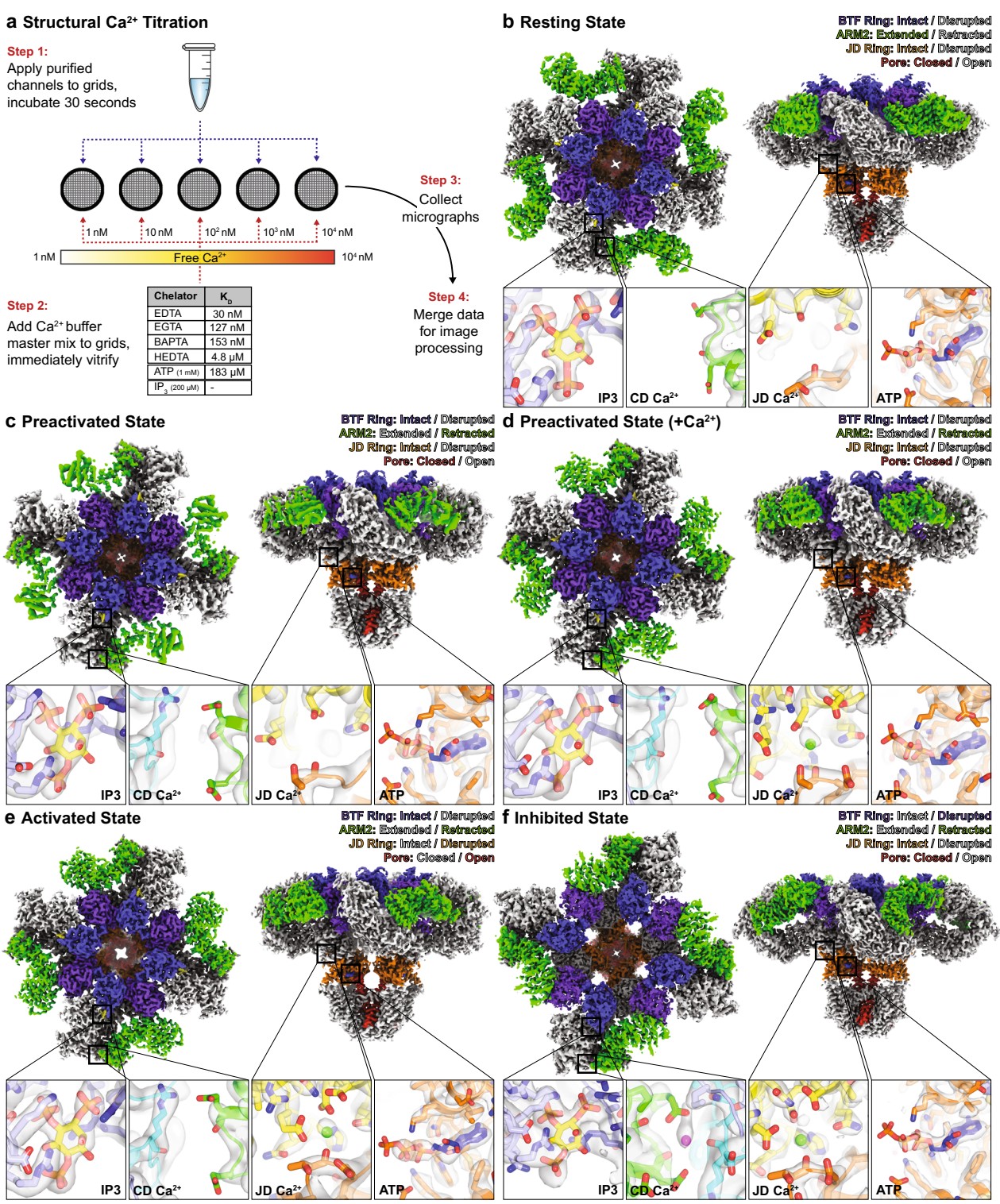

**Fig. 1 | Structural Ca²⁺ titration of human IP₃R3. a** Schematic for cryo-EM Ca²⁺ titration of hIP₃R3. **b–f** C4-symmetrized composite cryo-EM density maps viewed from the cytosol (left) and the side (right) with structural heuristics (top-right corner) and ligand binding sites (bottom insets for IP₃, CD Ca²⁺, JD Ca²⁺, and ATP) for the resting (**b**), preactivated (**c**), preactivated+Ca²⁺ (**d**), activated (**e**), and inhibited states (**f**). Insets in b-f are colored by domain: BTF1 (purple), BTF2 (blue), ARM1 (light blue), CLD (cyan), ARM2 (green), ARM3 (yellow), JD (orange), and TMD (red). Insets in (**b**) are contoured at 4, 8, 9 and 4 σ thresholds for the IP₃, CD Ca²⁺, JD Ca²⁺, and ATP sites, respectively. Insets in (**c–e**) are contoured at 5, 7, 9 and 4 σ thresholds for the IP₃, CD Ca²⁺, JD Ca²⁺, and ATP sites, respectively. Insets in (**f**) are contoured at 1, 9, 15 and 6 σ thresholds for the IP₃, CD Ca²⁺, JD Ca²⁺, and ATP sites, respectively.

concentration used for vitrification was sufficient to saturate the nM-affinity binding site[38], and that asymmetry of the ARM2 conformations did not arise from substoichiometric IP₃ binding. The IP₃-binding site is best resolved in the resting state where Arg568 on ARM1 coordinates

the 1-phosphate of IP₃ conferring a specific orientation to IP₃ in this pocket as predicted by mutagenesis[39]. Arg266 and Arg270 on BTF2, and Arg503, Lys507, Arg510, and Lys569 on ARM1 complete the positively charged binding site to coordinate IP₃ (PDB: 1N4K, 3T8S,

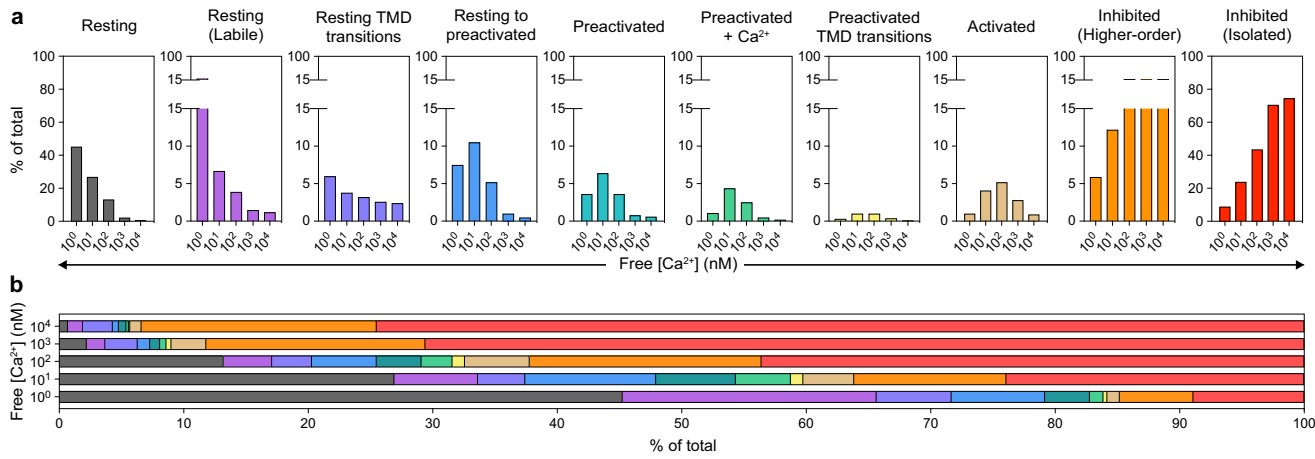

**Fig. 2 | Ca²⁺-dependent conformational landscape of hIP₃R3. a** Relative percent abundance of the five major states and the ensembles of minor states. **b** Aggregate abundances of all states across the Ca²⁺ titration. Graphs are colored by state: resting (grey), resting labile (magenta), resting TMD transitions (purple), resting-to-preactivated (blue), preactivated (cyan), preactivated+Ca²⁺ (green), preactivated TMD transitions (yellow), activated (sand), higher-order inhibited (orange), and isolated inhibited (red).

3UJ0)[40–42]. As observed previously[29], IP₃ can bind the channel via two modes (Supplementary Fig. 12). Comparing the resting state to a previously published ligand-free state (PDB: 6DQJ), IP₃ binding results in a contraction of the IP₃-binding pocket through movement of a loop (Leu265-Ser278) on BTF2 (Supplementary Fig. 12). Conversely, in the ARM2 retracted states, ARM1 tilts towards IP₃ to contract the ARM1-BTF2 interface. Notably, IP₃ is coordinated by the same residues in both binding modes (insets in Fig. 1b–f).

The Zn²⁺ ion bound in the JD is coordinated by a C₂H₂ zinc-finger fold formed by Cys2538, Cys2541, His2558, and His2563, where it has been observed in other IP₃R structures[28] (insets in Fig. 1b–f). The adenine base of the nearby ATP is buried in a hydrophobic cavity that was recently identified as an ATP-binding site that is structurally conserved with RyRs (Supplementary Fig. 13; PDB: 7T3P, 5TAP)[33,43]. Specificity for adenine bases[21,44–46] is imparted through the primary amine of the base forming interactions with the backbone carbonyl oxygen of His2558 and thiolate of Cys2538. The triphosphate moiety of ATP extends away from the JD with clear densities corresponding to the α and β phosphates, which are directly coordinated by Lys2152 and Lys2560, respectively (Fig. 1b–f). The γ-phosphate is poorly resolved and does not form direct interactions with the channel. Taken together, the coordination of ATP is consistent with both ATP and ADP having greater potentiating effects on IP₃Rs over AMP[21,44,45].

In contrast to the saturating conditions for IP₃ and ATP, our buffers sampled a range of Ca²⁺ concentrations that span the reported apparent affinities for both activation and inhibition of IP₃Rs, suggesting that we might resolve a range of Ca²⁺ occupancies among the major states. To assess the Ca²⁺-dependence of each conformation, we first inspected the cryo-EM density near the previously identified JD and CD Ca²⁺ binding sites[29] (Fig. 1b–f). In both the resting and preactivated states, no density peaks consistent with a bound Ca²⁺ ion were observed at either the JD or CD binding sites (insets in Fig. 1b–c). In the preactivated+Ca²⁺ state, we observed a density peak that we assigned as a Ca²⁺ in the JD site while the CD site was unoccupied (inset in Fig. 1d). The Ca²⁺-binding profile of the activated state is the same as the preactivated+Ca²⁺ state, with an occupied JD site and an empty CD site (inset in Fig. 1e). Only in the inhibited state did we observe densities corresponding to Ca²⁺ in both sites (inset in Fig. 1f). In the three JD Ca²⁺-bound states, the backbone of Thr2581 from the JD and side chains of Glu1882, Glu1946, and Gln1949 from ARM3 coordinate Ca²⁺ (Fig. 1d–f and below). The CD Ca²⁺, observed exclusively in the inhibited state, is coordinated by the backbone of Arg743 from the CLD and side chain of Glu1125 and backbone of Glu1122 from ARM2 (Fig. 1f and

below). Outside of the CD and JD sites, no densities consistent with bound Ca²⁺ ions could be identified in any of the maps. Taken together with our previous analyses of hIP₃R3 in saturating Ca²⁺[29], these data are consistent with the JD and CD sites being the primary Ca²⁺ binding sites in IP₃Rs. Thus, in addition to their distinct global conformations, the five major states display defining ligand-binding properties. The resting and preactivated states, which bind IP₃, ATP, and Zn²⁺, but not Ca²⁺, differ in how they coordinate IP₃. In addition to IP₃, ATP, and Zn²⁺, a single Ca²⁺ ion is bound to each protomer of the preactivated+Ca²⁺ and activated states, while two Ca²⁺ ions are bound to each protomer of the inhibited state.

## Ca²⁺ perturbs the energetic landscape of hIP₃R3

Single-particle cryo-EM analysis of vitrified samples represents a near equilibrium assessment of their conformational landscape, allowing one to infer relative conformational free energy from the number of particles that populate specific structural classes[47]. Therefore, by analyzing the effects of Ca²⁺ on the relative abundance of each hIP₃R3 conformation or ensemble, we can assess how Ca²⁺ biases the energetic landscape of the channel to favor activation at intermediate concentrations and favor inhibition at high concentrations (Fig. 2 and Supplementary Table 7). Furthermore, the Ca²⁺-dependent conformational landscape can provide additional confidence in the assignment of functional correlates to the observed conformational states. For example, the abundance of the putative resting state, which closely resembles the ligand-free state and shows no evidence of bound Ca²⁺ ions, is negatively correlated with the concentration of Ca²⁺. At low Ca²⁺, 45.2% of the particles adopt the resting state whereas this percentage drops to 0.7% at high Ca²⁺. Together, the two labile resting states follow a similar pattern, starting at 20.4% of the particles at 1 nM and falling to 1.2% at 10 μM. The ensemble of resting TMD transitions, comprised of the -C2 and -C4 states, is also similar, starting at 6% at 1 nM and falling to 2.4% at 10 μM.

We observed two distinct inhibited states – an isolated inhibited state and an assembled inhibited state in which several inhibited tetramers form higher-order assemblies (Fig. 1f and below, Supplementary Figs. 9 and 10). Although the states are structurally very similar with disrupted BTF rings, they have distinct abundance profiles with respect to Ca²⁺ concentration (Fig. 2a). The abundance of the isolated inhibited channels is the inverse of the resting state i.e. positively correlated to Ca²⁺ concentration, increasing monotonically to a maximum of 74.5% at 10 μM. The assembled inhibited state follows the same pattern at low Ca²⁺ concentrations, increasing from 5.9% at 1 nM

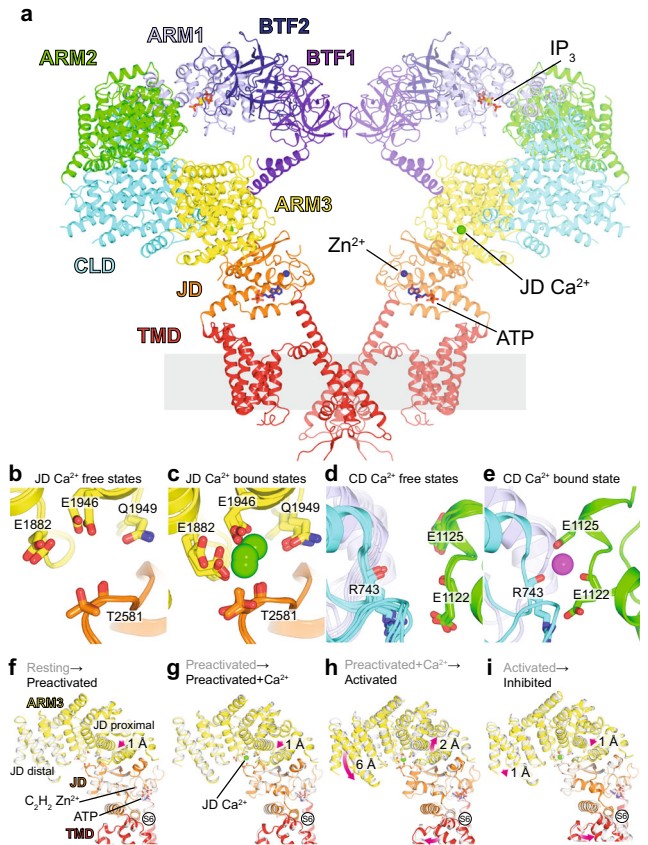

**Fig. 3 | $Ca^{2+}$ binding to the JD site has diverse effects on channel conformation.**
**a** Side view of the activated state highlighting domain architecture on the left protomer and ligand binding sites on the right protomer. Front and rear protomers removed for clarity. **b**–**c** Superpositions of the JD $Ca^{2+}$ binding site in the $Ca^{2+}$-free states (**b**) and $Ca^{2+}$-bound states (**c**). **d**–**e** Superpositions of the CD $Ca^{2+}$ binding site in the $Ca^{2+}$-free states (**d**) and the $Ca^{2+}$-bound state (**e**). **f**–**i** Superpositions of the ARM3-JD interface aligned by the JD for transitions from resting to preactivated (**f**), preactivated to preactivated+$Ca^{2+}$ (**g**), preactivated+$Ca^{2+}$ to activated (**h**), and activated to inhibited (**i**). Magenta arrows highlight movements of the proximal and distal regions of the JD between states.

to a maximum of 20.1% at 100 nM. However, higher $Ca^{2+}$ concentrations do not have any additional effect as the abundance of the assembled inhibited state plateaus between 17.6% and 20.1%. Although the structures of the tetramers in the higher-order assemblies are very similar to the isolated inhibited tetramers, their divergent $Ca^{2+}$-dependence suggests that they are distinct states and that formation of higher-order assemblies may represent an alternative mechanism for achieving an inhibited state, as we will discuss later.

In contrast to the resting-like states and the inhibited states, the distribution of the preactivated-like and activated states exhibit biphasic $Ca^{2+}$ dependencies, achieving their maximum abundance at intermediate $Ca^{2+}$ concentrations (Fig. 2a). Starting with the ensemble of resting-to-preactivated transitions, which achieve a maximum of 10.5% at 10 nM, the profiles of the preactivated, preactivated+$Ca^{2+}$, the ensemble of -C2 and -C4 preactivated TMD transitions, and the activated state are shifted rightward to progressively higher $Ca^{2+}$ concentrations. Apart from the activated state, the maximum abundance achieved by these states also decreases in a progressive manner, consistent with these states being progressively higher energy intermediates along a reaction coordinate extending from the resting state to the activated state. This continuum of inter-convertible states also provides a rationale for why the ensemble of resting-to-preactivated transitions and the preactivated state display a clear correlation with $Ca^{2+}$ despite not showing evidence of binding $Ca^{2+}$ themselves.

The abundance profile of the activated state agrees with decades of single-channel electrophysiological analyses of $IP_3Rs$, showing a biphasic open probability in the presence of saturating $IP_3$ and ATP with maximal activity occurring in the high nM $Ca^{2+}$ range (Fig. 2a)[19]. Moreover, the $Ca^{2+}$-dependent conformational landscape of $IP_3Rs$ resolves a bipartite mechanism for this biphasic relationship between $Ca^{2+}$ concentration and channel open probability. At low $Ca^{2+}$ $IP_3Rs$ must escape a low-energy ARM2 extended resting state in order to activate by binding $Ca^{2+}$ at the high-affinity JD site. At high $Ca^{2+}$, $IP_3Rs$ preferentially enter a low-energy inhibited state stabilized by a second $Ca^{2+}$ ion binding to the low-affinity CD site.

**The JD $Ca^{2+}$ site is essential for $Ca^{2+}$ oscillations**

The multimodal regulation of $IP_3Rs$, including activation and feedback inhibition by $Ca^{2+}$, produces $IP_3R$-dependent $Ca^{2+}$ oscillations in cells[48–51]. Structurally, we observe that $Ca^{2+}$ binding at the JD can occur in the putative activated state, while $Ca^{2+}$ binding at the CD site occurs only in the inhibited states (Fig. 3a–e). To assess the roles of these sites in producing cellular $Ca^{2+}$ oscillations and to attempt to establish a functional corollary to the conformational states obtained through the structural $Ca^{2+}$ titration, we employed a fluorescence-based $Ca^{2+}$ imaging assay that monitors $Ca^{2+}$ oscillations in cells. We first incubated HEK293T cells lacking all three $IP_3R$ isoforms ($IP_3R$-null) with Cal-520-AM, a fluorogenic calcium-sensitive dye, and then stimulated intracellular $IP_3$ generation by adding carbachol to the bath solution (Fig. 4a)[52]. Saturating carbachol concentrations (100 μM) were added to cells to minimize potential stimulus-dependent effects on the $IP_3R$ response in cells[53]. Consistent with earlier reports[52], no detectable changes in cytosolic $Ca^{2+}$ were observed in $IP_3R$-null cells (Supplementary Fig. 14). Conversely, $Ca^{2+}$ oscillations of two or more peaks were observed in cells transiently expressing $hIP_3R3$, indicating that the construct used for structural analysis expresses a functional channel (Fig. 4b). We assessed the temporal characteristics of the carbachol-stimulated $Ca^{2+}$ spikes in cells by aligning the initial peak of each normalized cellular trace that produced an oscillatory response (Fig. 4c). For $IP_3R$-null transiently expressing wild-type $hIP_3R3$, the mean slope of the rising phase at the half-maximal intensity was $0.103 \pm 0.015$ Fluorescence$_{norm}$ sec$^{-1}$. Traces were also analyzed to determine the number of peaks observed in cells showing oscillatory responses following carbachol stimulation, with cells expressing wild-type $hIP_3R3$ having a median of 4 peaks/cell (Fig. 4d). Finally, to calculate the time between successive $Ca^{2+}$ spikes (inter-spike interval), we extracted traces from segmented cells, then smoothed and adjusted the baseline to automatically identify peaks. For wild-type $hIP_3R3$ the mean inter-spike interval was 21.7 seconds, which is within the range of times measured for endogenous $IP_3R$-mediated cytosolic $Ca^{2+}$ [54,55].

Having established metrics that describe the carbachol-induced $Ca^{2+}$ oscillations of wild-type $hIP_3R3$, we next examined the effects of perturbing the $Ca^{2+}$-binding sites. We transiently expressed $hIP_3R3$ with mutations to the JD site (Glu1882Gln+Glu1946Gln), the CD site (Glu1125Gln) or both sites (Glu1125Gln+Glu1882Gln+Glu1946Gln) in $IP_3R$-null cells. Robust $Ca^{2+}$ oscillations were observed in cells expressing the CD mutant (Fig. 4e–g). While the mean rising phase was similar to wild-type $hIP_3R3$ (Fig. 3f, Supplementary Table 8), the mean inter-spike interval was nearly half (59%) at 12.7 seconds (Supplementary Fig. 14, Supplementary Table 8), suggesting that perturbing the CD site alters gating of $hIP_3R3$. As the CD site is exclusively occupied in the inhibited states, our structural and functional analyses are consistent with $Ca^{2+}$ binding at the CD site contributing to channel inhibition.

Unlike cells expressing wild-type channels or the CD mutant, we did not observe oscillatory responses in cells expressing either the JD mutant (Fig. 4h–j) or the JD/CD double mutant (Supplementary Fig. 14). Instead, we observed a single slow non-oscillatory event in both mutants that did not resemble the events seen in cells expressing

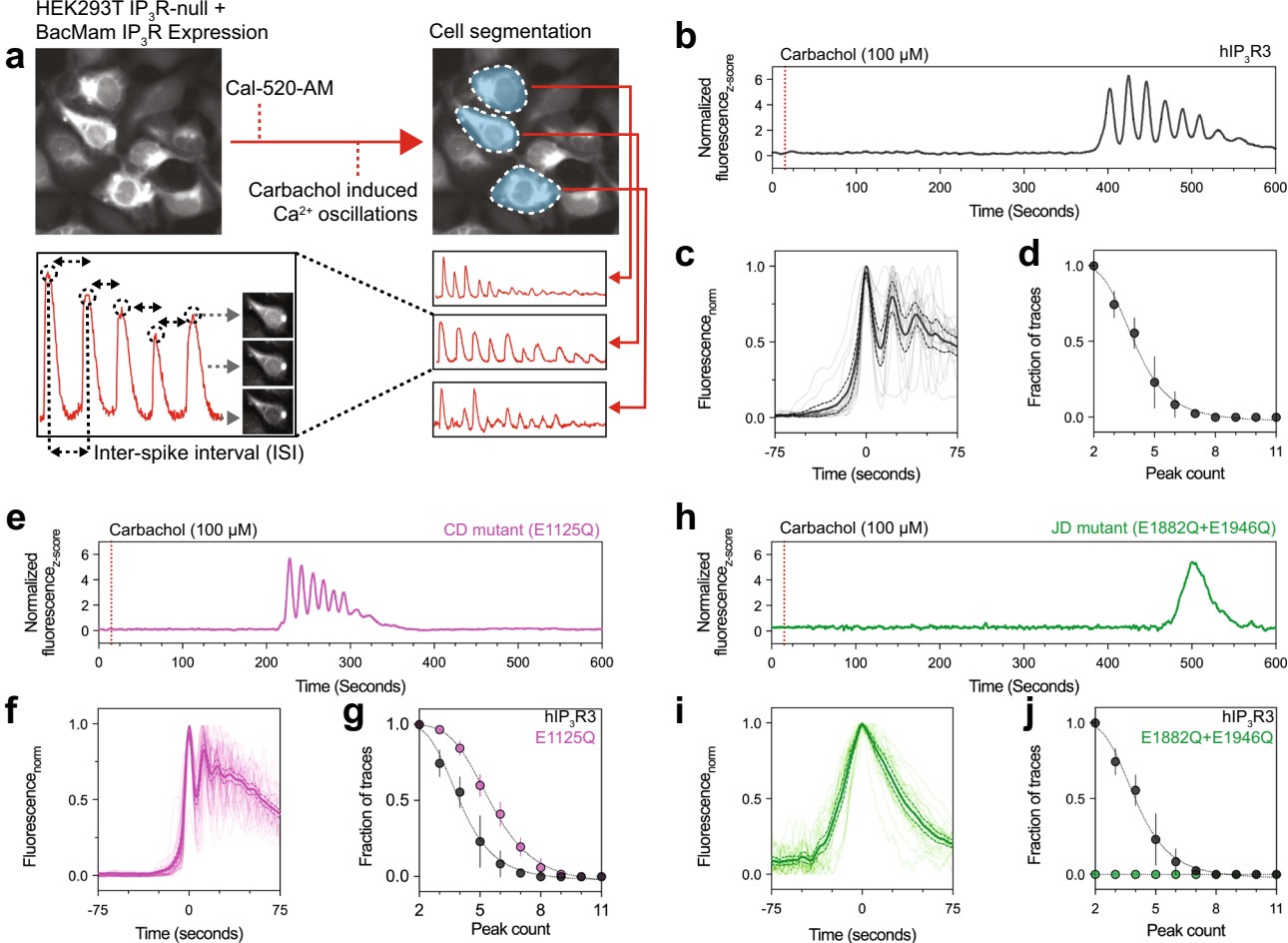

**Fig. 4 | Ca²⁺ binding to the JD site is required for Ca²⁺ oscillations. a** Schematic describing Cal-520-AM fluorescence-based Ca²⁺ imaging assay and data analysis. **b, e, h** Representative z-score normalized Cal-520-AM fluorescence traces recorded from cells expressing hIP₃R3 (**b**), CD mutant (**c**, E1125Q) and JD mutant (**h**, E1882Q +E1946Q) in an IP₃R-null background following stimulation by carbachol. $N = 3$ biologically independent samples. **c, f, i** Aligned first peak of every oscillatory trace (thin lines) normalized to 1 for hIP₃R3 (**c**), CD mutant (**f**, E1125Q) and JD mutant (**i**, E1882Q+E1946Q). Bold line represents mean and dashed lines represent 95% confidence interval. **d, g, j**, Peak count distributions for all oscillatory traces observed for hIP₃R3 (**d**), CD mutant (**g**, E1125Q) and JD mutant (**j**, E1882Q+E1946Q). Individual points represent mean and error bars represent S.E.M. Source data are provided as a Source Data file.

the wild-type channel. The mean slope of the rising phase was 3.7 times slower for cells expressing the JD mutant and 3.0 times slower for cells expressing the JD/CD double mutant than those of cells expressing wild-type hIP₃R3 (Fig. 3c, i and Supplementary Fig. 14). Therefore, although perturbations to the JD site do not abolish IP₃R-mediated Ca²⁺ release, consistent with recent electrophysiological analyses showing diminished activity of JD site mutants[56], the JD site is essential for ensuring the fidelity of agonist-evoked cytosolic Ca²⁺ oscillations in cells.

**Binding of the JD Ca²⁺ ion has distinct effects on channel conformation**

Although Ca²⁺ binding to the JD site is required for Ca²⁺ oscillations in cells, it is also occupied in the closed preactivated+Ca²⁺ and the closed inhibited states. To gain insights into how Ca²⁺ binding can stabilize these three distinct conformations, we aligned the JD of the five major states (Fig. 3f–i) to visualize the progressive changes to the JD Ca²⁺ binding site during activation and inhibition. The pairwise comparisons reveal that large changes to the ARM3-JD interface occur exclusively during the transition from the preactivated +Ca²⁺ to activated state: the JD-distal region of ARM3 rotates 6 Å towards the JD while the JD-proximal region shifts upwards 2 Å back to its resting state position (Fig. 3h). The changes that occur during the other transitions are more subtle. For example, the transitions

from resting to preactivated and from preactivated to preactivated +Ca²⁺ are each accompanied by 1 Å downward movements of the JD-proximal part of ARM3 (Fig. 3f–g). Binding a second Ca²⁺ at the CD site also results in a minimal rearrangement of the ARM3-JD interface, with both the distal and proximal regions of ARM3 moving down 1 Å during the transition from the activated to inhibited state (Fig. 3i). Surprisingly, despite the large global conformational differences between the preactivated+Ca²⁺, activated and inhibited states, the configuration of the residues that form the JD site are nearly identical. The JD binding site appears to adopt only two conformations, a Ca²⁺-free expanded conformation in the resting and preactivated states and a Ca²⁺-bound contracted conformation in the preactivated+Ca²⁺, activated and inhibited states (Fig. 3d–e). Furthermore, we only observe stable occupancy of the JD site in the ARM2 retracted states, suggesting that the IP₃-stabilized movement of ARM2 increases the affinity for Ca²⁺. Allosteric coupling between Ca²⁺ and IP₃ binding is consistent with biochemical experiments suggesting that Ca²⁺ binding can increase the affinity for IP₃[57,58], and kinetic experiments showing IP₃ binding exposes a high-affinity Ca²⁺ binding site[59]. In summary, although Ca²⁺ binding to the JD site stabilizes a single, distinct Ca²⁺-bound conformation of the binding site, the effect of Ca²⁺ binding on channel conformation at the global level can be varied and is influenced by the presence of other ligands.

## IP$_3$ primes channel activation through a cooperative process involving ARM2 retraction

Activation of IP$_3$Rs requires that all four IP$_3$ binding sites be intact[52], suggesting that a coordinated IP$_3$-mediated conformational change must occur prior to pore opening. Our previous analysis revealed that the transition between ARM2 extended and ARM2 retracted states is both IP$_3$-mediated, with the retracted state only being resolved in the presence of IP$_3$, and cooperative, with the four-fold symmetric extended or retracted conformations being substantially favored over the asymmetric states as opposed to a binomial distribution[29]. We therefore hypothesized that the IP$_3$ binding mode of a protomer can be sensed by its neighbors and that this communication may underlie the requirement for four intact IP$_3$ binding sites. To evaluate the relationships between a single protomer and its neighbors, we performed symmetry expansion, focused refinement, and 3DVA on the CD of a single protomer, which includes the uniformly occupied IP$_3$ binding site and ARM2, for the resting-to-preactivated ensemble (Fig. 5a and Supplementary Table 4). By calculating reconstructions for particles segmented along the primary dimension of variability, we can visualize the progression of one protomer (labeled b in Fig. 5b–g) from the ARM2 extended conformation resolved in the resting state to the ARM2 retracted conformation of the preactivated state. In the most extended ARM2 position of the central protomer, ARM2$^b$ forms two interactions with the counterclockwise protomer (labeled a), one with ARM1$^a$, and a second with BTF1$^a$ (Fig. 5b). The transition of protomer $a$ to the ARM2 retracted state is accompanied by a contraction of the ARM1-BTF2 interface around IP$_3$. A consequence of this contraction is that ARM1$^a$ is pulled away from ARM2$^b$, disrupting one of ARM2$^b$'s interprotomer interactions (Fig. 5c). The diminished association with the neighboring protomer results in a more dynamic state for ARM2$^b$, which manifests in weaker averaged density at its distal end (Fig. 5d). The increased flexibility of ARM2$^b$ destabilizes its remaining interprotomer interaction with BTF1$^a$ and allows it to transiently disengage from BTF1$^a$ and rotate towards CLD$^b$ to adopt the retracted conformation. In the retracted conformation, ARM2$^b$ establishes a new interprotomer interface with BTF1$^a$ (Fig. 5e). ARM2$^b$ retraction results in a tilt of ARM1$^b$ away from ARM2 on the clockwise protomer and the entire progression repeats, enabling a cascade around the tetramer that primes the JD site for Ca$^{2+}$ binding (Fig. 5f–g).

The observed continuum from a symmetric ARM2 extended resting state to a symmetric ARM2 retracted preactivated state suggests that this process is reversible despite the presence of saturating IP$_3$. Consistent with the process being reversible, more particles adopt the resting state than do the ARM2 retracted preactivated and preactivated+Ca$^{2+}$ states (Fig. 2). Potentially contributing to the favorability of the ARM2 extended state is a loop between Pro897 and Glu958 of the CLD, which we call the wedge loop. In the resting state, a portion of the wedge loop, including Thr926-Ala943, inserts into a cavity surrounded by the CLD, ARM1, ARM2 and ARM3 and adopts an ordered conformation (Fig. 5h–i). Compared to the resting state, ARM2 retraction in the preactivated, preactivated+Ca$^{2+}$, activated and inhibited states is accompanied by a contraction of this cavity. Modeling the resting state conformation of Thr926-Ala943 into the ARM2 retracted states, where we observed no density for the wedge loop, reveals several steric clashes that would likely disfavor binding of the wedge loop (Supplementary Fig. 15 and Supplementary Table 6).

To assess the relationship between ARM2 retraction and wedge loop binding, we recalculated the ARM2 extended portion of the 3DVA trajectory for the resting-to-preactivated transitions with finer sampling. By aligning the maps based on the strength of the density for the wedge loop, we found that the flexibility of ARM2, as assessed by the local quality of the density, is inversely correlated with the strength of the wedge loop density, indicating that the presence of the wedge loop stabilizes ARM2 in the extended conformation (Supplementary Fig. 15). Moreover, this alignment reveals how the wedge loop

dissociates from its binding site in a stepwise fashion. First to dissociate are the residues surrounding Arg931, followed by the N- and C-terminal ends of the loop. Phe936 is the last residue to become disordered, indicating that Phe936 is critical for the interaction.

Flanking Phe936 is the conserved residue Ser934, which can be phosphorylated by protein kinase A (Fig. 5i)[60–62]. Mutation of the residue homologous to Ser934 in hIP$_3$R2 to alanine abrogates the ability of protein kinase A to sensitize hIP$_3$R2 to low-level stimulation by carbachol[63]. Modeling in a phosphorylated serine at position 934 places the phosphate group in close proximity to Ser937, potentially destabilizing the conformation of the wedge loop and weakening the critical interactions formed by Phe936, suggesting that phosphorylation of Ser934 may influence channel activity by destabilizing the resting state. The residues on and around the wedge loop described here are conserved among the three human IP$_3$R isoforms, suggesting that the wedge loop may serve as a conserved regulatory motif that can influence the equilibrium between ARM2 extension and retraction and thus alter the affinity of the JD site for Ca$^{2+}$ in all IP$_3$Rs (Supplementary Figs. 10 and 15).

To explore the role of the ARM2-mediated conformational changes in channel activation, we deleted the ARM2 domain (dARM2 mutant; Ala1101-Trp1586) and assessed the effects of its loss on Ca$^{2+}$ oscillations (Fig. 5j–l). Compared to cells expressing wild-type hIP$_3$R3, carbachol stimulated Ca$^{2+}$ oscillations were observed less frequently ($n_{WT}$ = 74; $n_{dARM2}$ = 14) in cells expressing the dARM2 mutant despite both being expressed in a similar fraction of cells (Fig. 5l). Also diminished was the frequency of the Ca$^{2+}$ spikes. The inter-spike interval was on average 4.7 times longer in cells expressing the dARM2 mutant than in cells expressing hIP$_3$R3. Although the Ca$^{2+}$ spikes were infrequent, the mean slope of the rising phase of the few responding cells was similar to that of cells expressing wild-type hIP$_3$R3, suggesting that the dARM2 mutant is functional. Thus, while ARM2 is not required for activation or inhibition, its loss appears to reduce the likelihood of exceeding the threshold required for Ca$^{2+}$ wave propagation[50,64]. Together, our structural and functional analyses reveal that IP$_3$ binding favors adoption of the preactivated state, which displays a higher apparent affinity for Ca$^{2+}$, despite the presence of several channel intrinsic features that favor the resting state.

## Activation of hIP$_3$R3 by IP$_3$, Ca$^{2+}$ and ATP

Comparing the closed, Ca$^{2+}$-bound preactivated+Ca$^{2+}$ state with the open, Ca$^{2+}$-bound activated state allows us to observe the conformational changes that enable the pore to open. Although both states share a common ligand-binding profile, large conformational changes can be observed extending from the JD Ca$^{2+}$ binding site through the JD to the TMD (Fig. 3). The JD, which connects the ligand-binding sites in the cytosolic domain to the TMD, is composed of two interwoven, discontinuous segments of the polypeptide that connect to both the N- and C-terminal ends of the TMD. Together, these connections enable the JD to alter the conformation of both domains of the TMD: the central pore and the peripheral S1-S4 domain (Fig. 6f, h–i). In the closed states, the four JDs assemble into a tetrameric ring structure. In the activated state, the contraction of the ARM3-JD interface induces a -13° clockwise rigid-body rotation of the JDs that disrupts the inter-JD interactions (Fig. 6g).

The first segment of the JD, which we call JD-A (Glu2111-Met2191), is connected to the peripheral S1-S4 domain. The pore domain and the domain-swapped S1-S4 domain are connected by a short amphipathic S4-S5 linker helix (Fig. 6f). In the closed states, the S4-S5 linkers form a belt around the S6 helices that stabilize the closed pore. In the activated state, the rotation of the JD tilts S1-S4 towards the luminal side of the membrane, relaxing the belt around S6 (Fig. 6a–f and Supplementary Movie 11).

The second segment of the JD, which we call JD-B (Cys2538-Met2608), is directly linked to the cytosolic end of the pore-lining S6

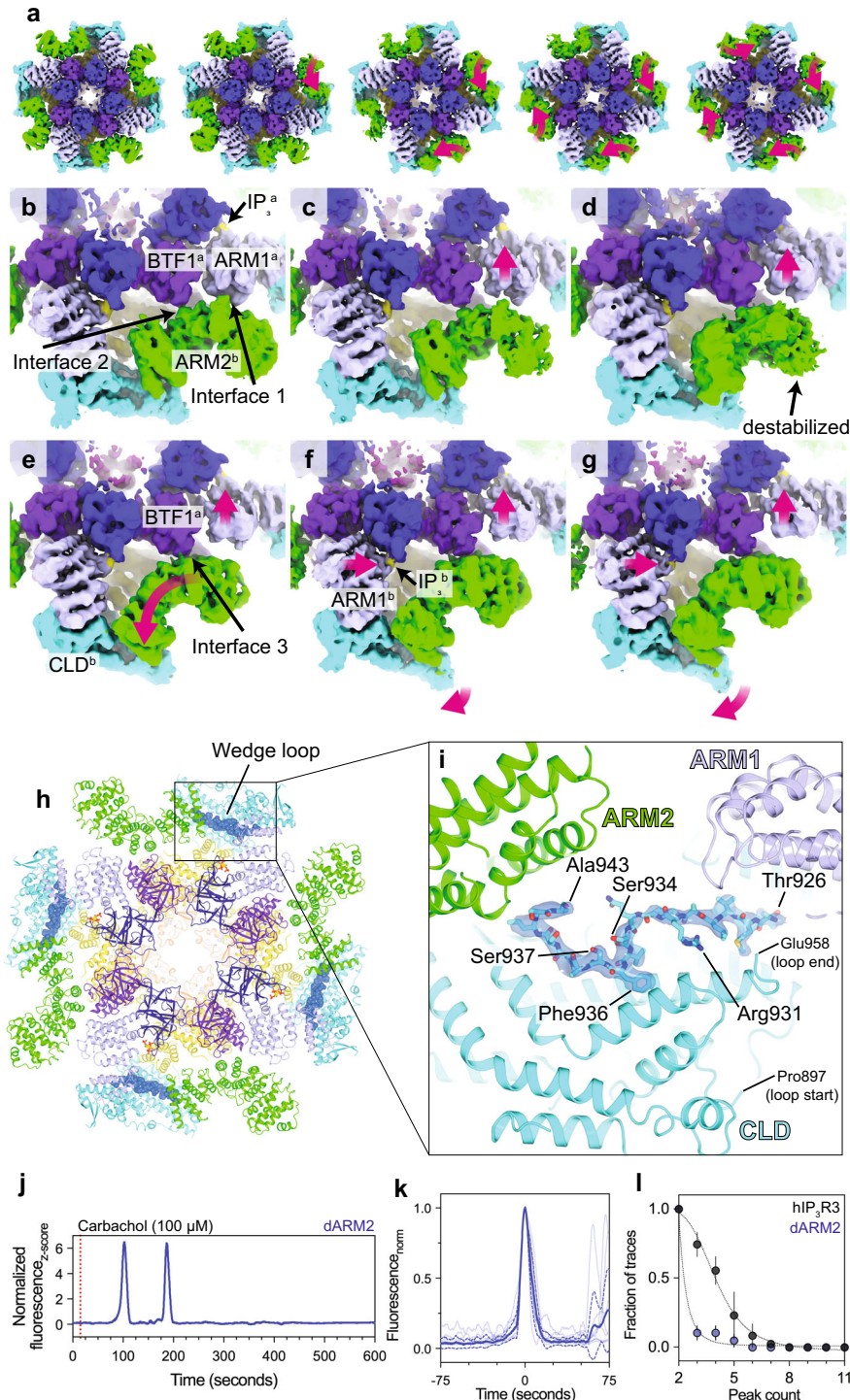

**Fig. 5 | IP$_3$ primes the channel for activation via a cooperative process involving ARM2. a** Unsharpened 5 Å low-pass filtered cryo-EM density of five states showing a range of ARM2 positions between the resting state (left) and preactivated state (right). The three intermediates are derived from the resting-to-preactivated transitions (see Supplementary Table 4). Magenta arrows highlight movements of ARM2 compared to the preceding panel. **b**–**g** Unsharpened 5 Å low-pass filtered cryo-EM density along a trajectory of protomer b from the extended state to the retracted state (see Supplementary Table 4). **b** ARM2$^b$ forms two interactions with the adjacent protomer in the extended state. IP$_3$ bound to adjacent protomer a is highlighted. **c** The first movement is the displacement of ARM1$^a$ away from ARM2$^b$. **d** Further displacement of ARM1$^a$ away is accompanied by a destabilization of the distal end of ARM2$^b$. **e** ARM2$^b$ is repositioned into the retracted conformation near CLD$^b$ where ARM2$^b$ can contact BTF1$^a$ as ARM2$^a$ continues to move towards IP$_3$$^a$. **f** Once ARM2$^b$ adopts the retracted conformation, ARM1$^b$ can move towards the

bound IP$_3$ of protomer b, repeating the progression. This process results in torsion of the CLD$^b$. **g** The movements reach their extremes in the retracted conformation. **h** Resting state shown as cartoon viewed from the cytosol with wedge loop shown as blue spheres. **i** The wedge loop occupies a cavity between ARM1, ARM2 and the CLD in the resting state. Ordered residues within the wedge loop are depicted as sticks. Cryo-EM density for the wedge loop is shown as a blue isosurface.
**j** Representative z-score normalized Cal-520-AM fluorescence trace recorded from cells expressing the dARM2 mutant in an IP$_3$R-null background following stimulation by carbachol. $N$ = 3 biologically independent samples. **k** Aligned first peak of every oscillatory trace (thin lines) normalized to 1. Bold lines represent mean and dashed lines represent 95% confidence interval. **l** Distribution of peak counts for all oscillatory traces. Individual points represent mean and error bars represent S.E.M. Source data are provided as a Source Data file.

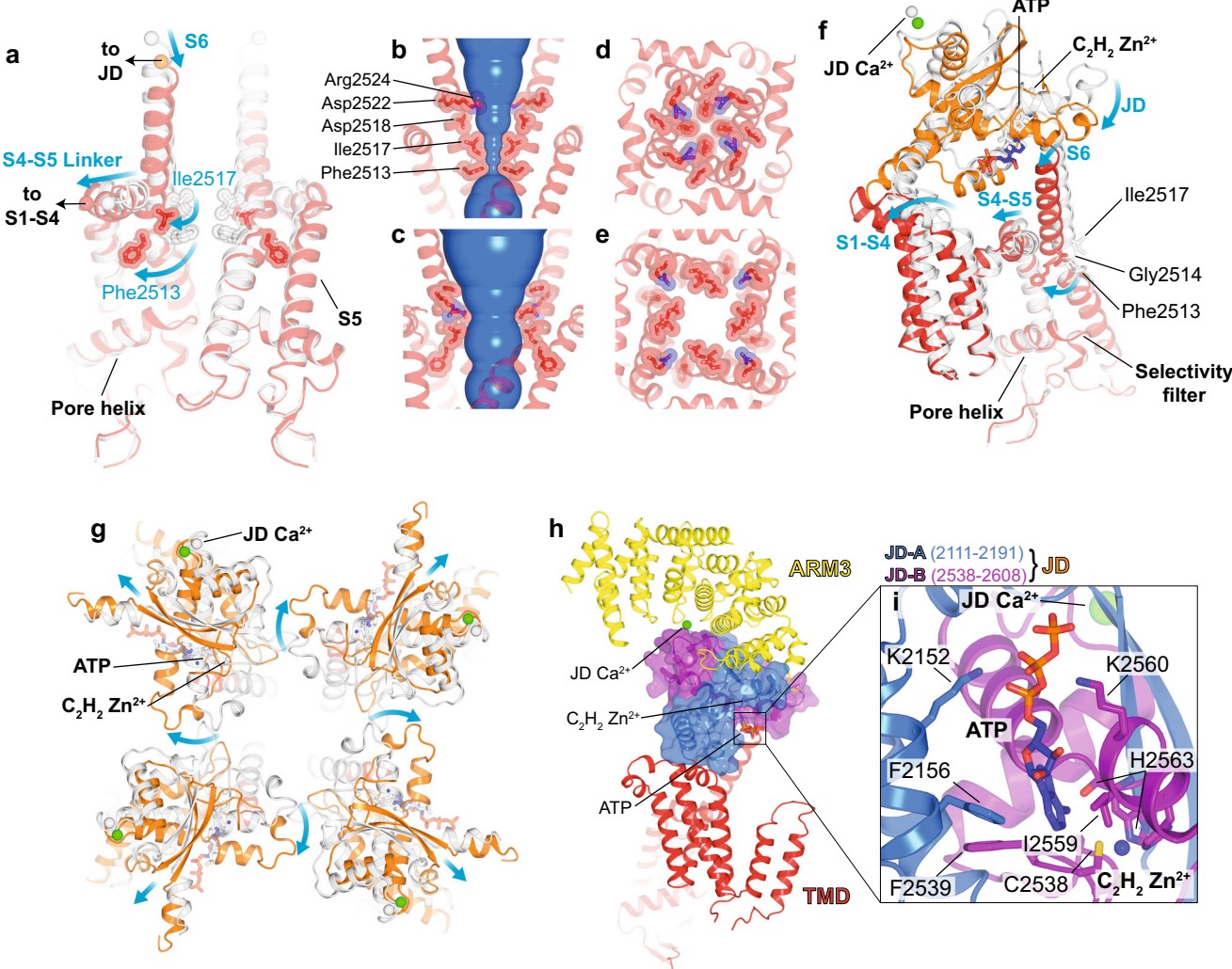

**Fig. 6 | Mechanism of activation. a** Superposition of the pore of the preactivated +Ca²⁺ (gray) and activated states (red), aligned by the luminal halves of S5 and S6, pore helix and selectivity filter. Front and rear protomers removed for clarity. Blue arrows highlight movement of S6, S4-S5 linker, and gating residues Phe2513 and Ile2517. Arrows show where the pore connects to S1-S4 domain and JD. **b**–**c** HOLE diagram showing solvent-accessible surface area of conduction pathway in preactivated+Ca²⁺ (**b**) and activated states (**c**). **d**–**e** Top view of constriction in preactivated+Ca²⁺ (**d**) and activated (**e**) states. **f** Comparison of TMD and JD of a single protomer of preactivated+Ca²⁺ (gray) and activated (colored) states aligned as in a. Blue arrows highlight the movements of the JD, S1-S4 bundle, S6, and the S4-S5 linker. Bending and rotation of S6 occurs at Gly2514 enabling Phe2513 and Ile2517 to repack behind the pore. **g** Comparison of JD ring of preactivated+Ca²⁺ (gray) and activated (colored) states viewed from the cytosol and aligned as in a. Arrows depict the movements that result in JD ring disruption during activation. **h** The JD (shown here in the activated state) is composed of two fragments JD-A (blue) and JD-B (purple). It is positioned between ARM3 and the TMD, and contributes to the JD Ca²⁺, ATP, and Zn²⁺ binding sites. **i** Inset highlights the ATP and Zn²⁺ binding sites at the interface between JD-A and JD-B.

helix. In the activated state, rotation of the JD pulls the S6 helices away from the center of the pore, stabilizing a 13° bend and ~30° rotation about the helical axis of S6 with Gly2514 being the pivot for both. Together, the tilt and rotation of S6 reposition Phe2513 and Ile2517, which seal the pore in the closed states, out of the ion conduction pathway to create an open pore with a minimum radius of 4 Å (Fig. 6a–e, Supplementary Fig. 16 and Supplementary Movies 9 and 10). In addition to changing the dimensions of the pore, the tilt and rotation of S6 reorient the side chains of Arg2524, Asp2518 and Asp2522, switching the pore from an electropositive to electronegative environment that would be favorable for cation conductance (Supplementary Fig. 16).

Our structural analyses indicate that both segments of the JD contribute to pore opening, with the rotation of JD-A relaxing the belt around the S6 helices and JD-B pulling the S6 helices apart. At the interface between these segments is an ATP (Fig. 6h–i and Supplementary Fig. 13). The adenine moiety of ATP is nestled in a hydrophobic pocket between the two segments lined by Phe2156 from JD-A

and Phe2539 and Ile2559 from JD-B (Fig. 6i). The phosphate groups similarly bridge the two segments of the JD with the α-phosphate coordinated by Lys2152 of JD-A and the β-phosphate coordinated by Lys2560 from JD-B. In cells, where ADP and ATP are abundant and the binding site should be predominantly occupied, ADP and ATP likely serve as molecular glue to hold the two discontinuous segments of the JD together. In the absence of ADP or ATP, Ca²⁺ binding may yield uncoupled movements of the two segments that would be a barrier to opening the pore, consistent with the prevailing model for ATP potentiation through sensitizing the channel to Ca²⁺ activation without affecting maximal open probability or high-Ca²⁺ inhibition[19,21]. Supporting the critical role of a rigid JD domain in channel activation, even a single cysteine-to-serine mutation at the JD Zn²⁺ binding site results in a complete loss of function without diminishing protein expression or IP₃ affinity[65].

In addition to the fully-open activated state, our analysis identified several minor classes with partially expanded pores that may represent snapshots of the rearrangements that occur during pore opening

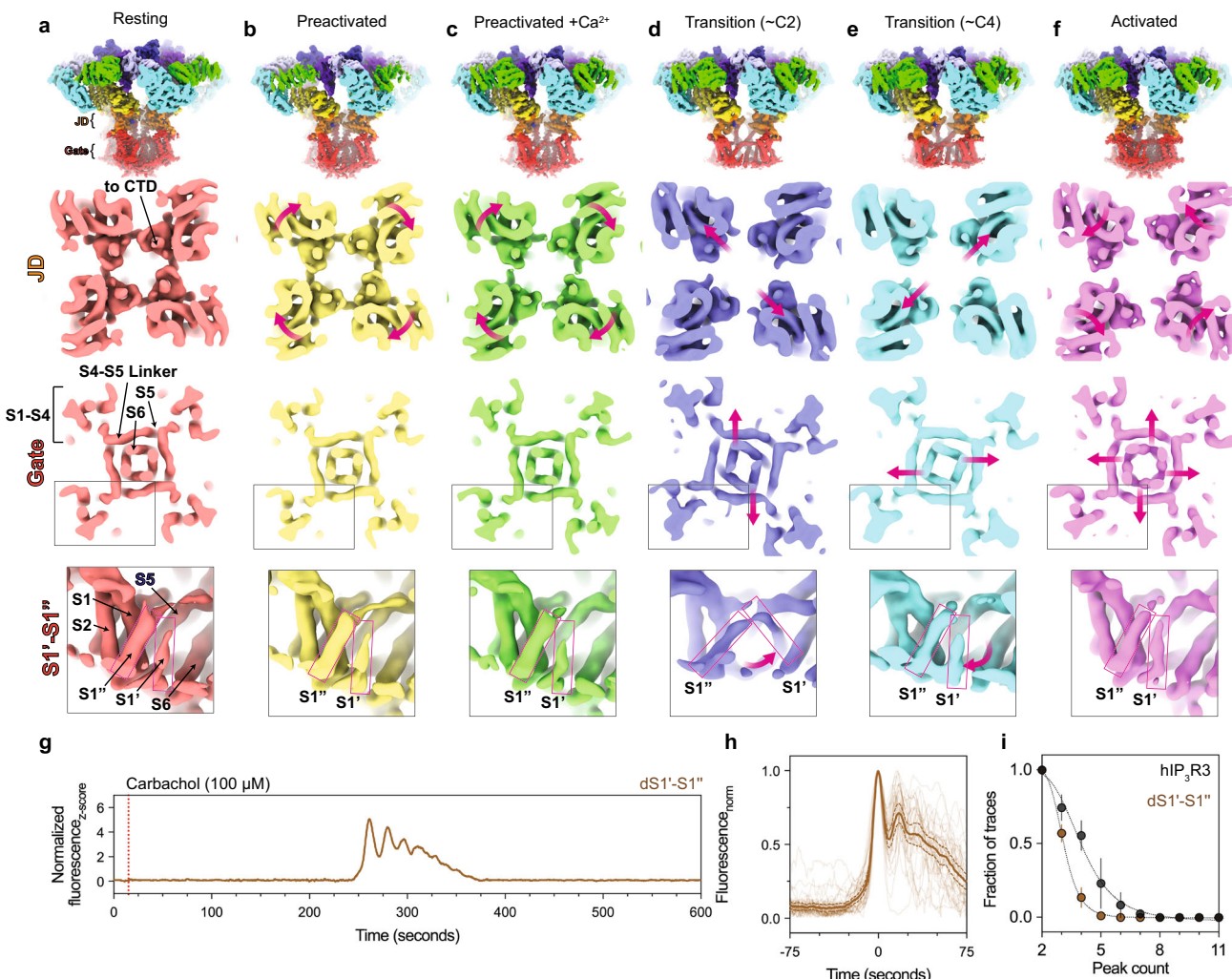

**Fig. 7 | Snapshots of the conformational rearrangements in the JD and TMD that enable gating. a–f** Cryo-EM density maps of the resting (**a**), preactivated (**b**), preactivated+$Ca^{2+}$ (**c**), -C2 preactivated TMD transition (**d**), -C4 preactivated TMD transition (**e**), activated (**f**), and inhibited (**g**) states, low-pass filtered to 4 Å (overall) or 7 Å (slices). Row 1: Overall cryo-EM density viewed from the side. Row 2: density slice looking from the cytosol at the height of the JD ring with magenta arrows highlighting movements of the JDs. Row 3: density slice looking from the cytosol at the height of the gate with magenta arrows highlighting movements of the S6

helices. Row 4: Side view of a single S1-S4 domain. **g** Representative z-score normalized Cal-520-AM fluorescence trace recorded from cells expressing the dS1'-S1" mutant in an IP3R-null background following stimulation by carbachol. $N = 3$ biologically independent samples. **h** Aligned first peak of every oscillatory trace (thin lines) normalized to 1. Bold line represents mean and dashed lines represent 95% confidence interval. **i** Distribution of peak counts for all oscillatory traces. Individual points represent mean and error bars represent S.E.M. Source data are provided as a Source Data file.

(Fig. 7d–e). While the local resolution near the pore of these reconstructions preclude atomic model building, comparing sections of the density maps can inform about how the pore and JDs move during gating. Two of these minor classes belong to the ensemble of preactivated TMD transitions. In one of the classes, two opposing pore-lining S6 helices tilt away from the axis of the pore, while the other two S6 helices are unchanged, establishing a -C2 pore configuration (Fig. 7d). The S4-S5 linkers also adopt a -C2 configuration with the S4-S5 linkers of the protomers with displaced S6 helices shifted away from the S6 helix of the neighboring protomer. This uncoupled S4-S5 linker conformation is consistent with our model that the S4-S5 linker belt must be relaxed for the pore to open and appears to be stabilized by an interaction with S1' of the adjacent protomer (inset in Fig. 7d). In the -C2 transition, the adjacent S1" tilts towards the pore allowing S1' to insert underneath the S4-S5 linker of the adjacent protomer, potentially stabilizing this intermediate state. To examine the role of the IP3R-specific S1' and S1" helices in channel gating, we monitored cytosolic $Ca^{2+}$ responses following carbachol stimulation in cells transduced with hIP3R3 lacking S1' and S1" (dS1'-S1") (Fig. 7g–i).

Compared to cells expressing wild-type hIP3R3, cells expressing dS1'-S1" had an increased inter-spike interval and decreased number of oscillatory events per cell, suggesting that S1' and S1" are required for faithful channel function. Intriguingly, while the analogous linkage between S1 and S2 is a poorly ordered acidic loop in the distantly-related RyRs[66], two helices preceding the S1-S4 domain occupy a position similar to S1'-S1" in IP3Rs[43]. Future studies will be necessary to uncover the role of this unique insertion between S1 and S2 in IP3Rs.

In the -C4 subclass, the S4-S5 linkers and the S6 helices of all four protomers are outwardly displaced, creating a partially dilated pore (Fig. 7e). However, compared to the activated state (Fig. 7f), the dilation appears to be incomplete as the cytosolic ends of S6 remain closer together. Comparing the JD in the preactivated+$Ca^{2+}$ and activated states reveals that the JD also adopts an intermediate conformation. Whereas the JDs are both separated and rotated in the activated state, the JDs in the -C4 transition are only separated.

Interpolating the -C2 and -C4 preactivated TMD transitions into a trajectory that begins with the resting state and ends with the activated state suggests a progression of JD rearrangements that facilitate gating

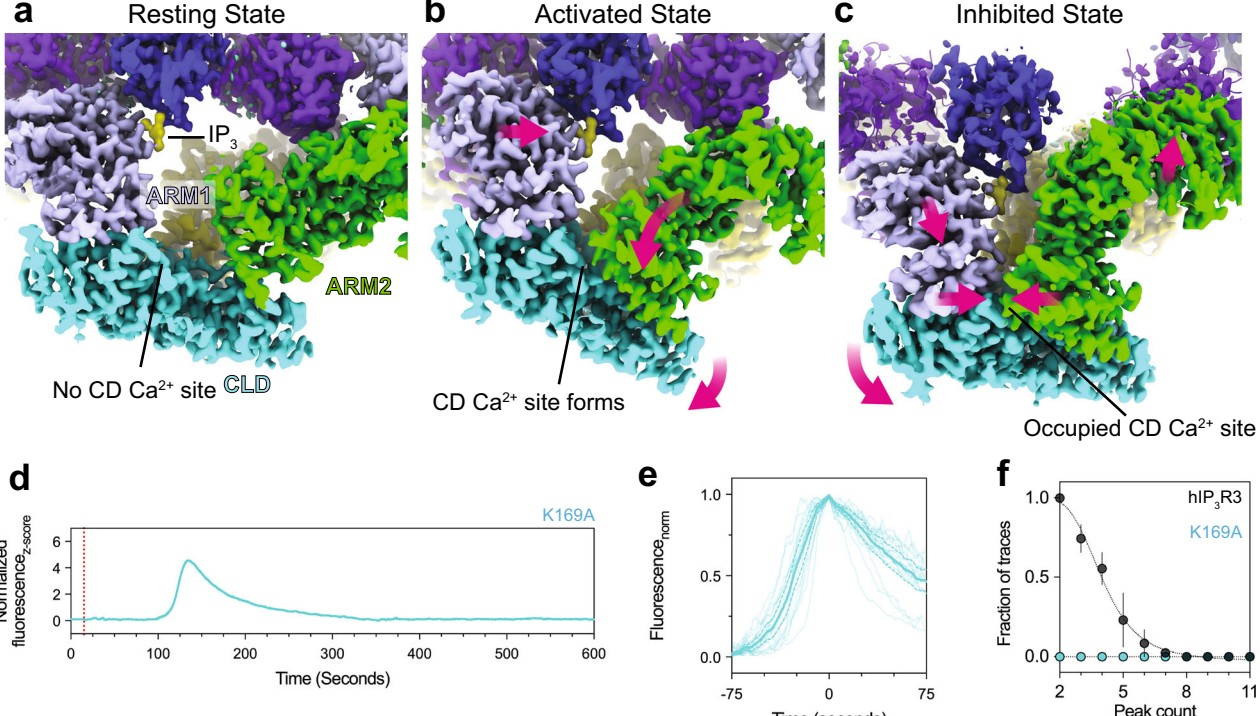

**Fig. 8 | Mechanism of high-Ca²⁺ inhibition. a–c** Cryo-EM density maps depicting the trajectory of a single protomer's CD from resting to activated to CD Ca²⁺-bound inhibited state. **a** In the resting state, the CD Ca²⁺ binding site does not exist because ARM2 is extended away from the CLD and ARM1. **b** ARM2 retraction creates the CD Ca²⁺ binding site in the preactivated, preactivated+Ca²⁺, and activated states, but no Ca²⁺ is yet bound. An interaction between ARM2 and BTF2 of the adjacent protomer restricts the movement of ARM2 and must be released to enable Ca²⁺ binding. **c** BTF ring disruption allows ARM2 to move further towards the CLD and bind the CD Ca²⁺

in the pore (Fig. 7a–f). First, retraction of ARM2 in the preactivated state results in a clockwise rotation of the JDs which is further magnified by Ca²⁺ binding in the preactivated+Ca²⁺ state. Once Ca²⁺ is bound, the channel can sample the ~C2 transition, where two opposing JDs shift outwards, disrupting the JD ring and loosening the S4-S5 linker belt leading to an outward movement of two of the four S6 helices. Then, the remaining two JDs are displaced away from the pore axis, resulting in a partial dilation of the pore. Finally, in the activated state, the JDs rotate about the helical axis of S6 to stabilize a fully-open pore where the hydrophobic gating residues Phe2513 and Ile2517 are repacked away from the permeation pathway. Notably, we do not observe any conformational changes in the pore helix or selectivity filter between the high-resolution closed and open states, indicating that the positions of Phe2513 and Ile2517 determine the gating state of the pore.

Subclasses with ~C2 and ~C4 distortions of the pore are also present in the ensemble of resting TMD transitions. In contrast to the preactivated TMD transitions, the JD ring remains intact in these subclasses, suggesting that the pore conformation is not strictly coupled to that of the JD ring (Supplementary Fig. 8). The structural association between the JD ring and TMD in IP₃Rs is thus weaker than the associations described between the pore and the cytosolic gating domains of other 6TM cation channels such as the BK channel (Slo1)[67].

### Mechanisms of high Ca²⁺ inhibition

Compared to the states with Ca²⁺ bound solely at the JD site, Ca²⁺ binding at the CD site in the inhibited state is accompanied by large conformational changes throughout the CD (Fig. 1b–f). The most prominent change is the disruption of the BTF ring, which results in the

ion. **d** Representative z-score normalized Cal-520-AM fluorescence trace recorded from cells expressing the K169A mutant in an IP₃R-null background following stimulation by carbachol. $N = 3$ biologically independent samples. **e** Aligned first peak of every oscillatory trace (thin lines) normalized to 1. Bold line represents mean and dashed lines represent 95% confidence interval. **f** Distribution of peak counts for all oscillatory traces. Individual points represent mean and error bars represent S.E.M. Source data are provided as a Source Data file.

CDs of the four protomers moving away from one another and towards the membrane. Despite employing the same classification approaches that resulted in identification of several other low-abundance intermediates, we did not observe any transition states between BTF ring intact and BTF ring disrupted states, suggesting that loss of a single interprotomer interaction may be sufficient to disrupt the BTF ring in a highly-cooperative fashion. Due to the presence of a second Ca²⁺ ion bound at the CD site, and because we previously demonstrated that BTF ring disruption insulates IP₃-mediated conformational changes from the channel gate[29], we hypothesized that this BTF ring-disrupted conformation is the high-Ca²⁺ inhibited state of the channel. Consistent with BTF ring disruption being a key aspect of inhibition, mutations at the interface between BTF1 and BTF2 of the neighboring protomer diminish or eliminate carbachol-induced Ca²⁺ oscillations in cells (Fig. 8d–f and Supplementary Fig. 14). Cells expressing a Trp168Ala/Lys169Ala double mutant displayed no detectable increase in cytoplasmic Ca²⁺ following carbachol stimulation, while only a single event could be observed in cells expressing a Lys169Ala mutant. These results corroborate mutagenesis experiments that predate structures of a full-length IP₃R that yielded a graded effect on IP₃-induced Ca²⁺ release from microsomes, with single mutations at the BTF1-BTF2 interface diminishing release compared to wild-type channels, and two or more mutations resulting in no detectable Ca²⁺ release[68].

Coordination of a Ca²⁺ in the CD site of the inhibited state is achieved by the N-terminal portion of the CLD and ARM2 rotating towards one another by a total of 3 Å compared to their positions in the activated state (Fig. 8b–c). Through ARM1, the rotation of the CLD pulls BTF1 and BTF2 outwards, away from the BTF domains of the neighboring protomers, while the rotation of ARM2 breaks its

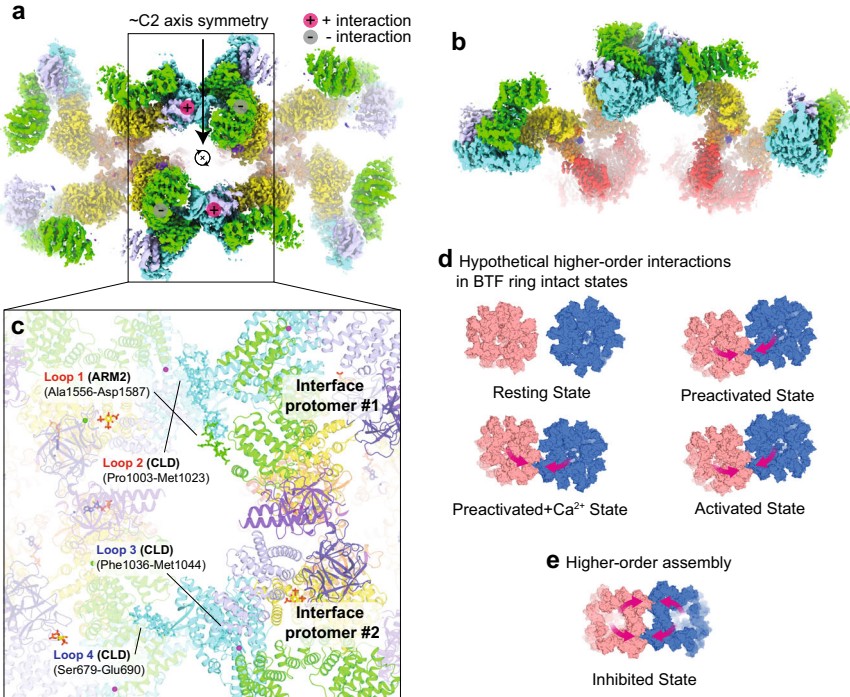

**Fig. 9 | hIP₃R3 can form higher-order assemblies. a, b** Top (**a**) and side (**b**) views of the octameric assembled inhibited state. **c** Atomic model showing the four interacting loops at the interface as boxed in a. For clarity, ordered loops are only shown for the tetramer at the right. **d** Surface representation of a modeling of hypothetical higher-order interactions for the resting, preactivated and activated states shows that steric restrictions imposed by the intact BTF ring allow only a single interaction to form between adjacent tetramers. **e** BTF ring disruption relieves this restriction and allows higher-order interactions to occur in a reciprocal fashion between adjacent tetramers. Arrows highlight the movements of two CDs that together establish the second inter-tetramer interface.

interaction with BTF1 of the neighboring protomer. From these observations, Ca²⁺ binding to the CD site stabilizes the BTF ring disrupted conformation. However, our data cannot discern if Ca²⁺ binding at the CD site is achieved through an induced fit mechanism or through conformational selection.

## Inhibited state particles can form higher-order assemblies

Classification identified two distinct populations of inhibited particles with disrupted BTF rings – individual tetramers and higher-order assemblies of two or more tetramers. The contacts between adjacent tetramers in the assemblies are mediated by two reciprocal interaction surfaces that together create a pseudo two-fold symmetry axis between the two tetramers. Within each of the tetramers, one protomer forms what we call the plus (+) interface through its CLD, one promoter forms what we call the minus (−) interface through its ARM2 domain while the remaining two protomers do not participate in the interaction (Fig. 9a–c). The + interface of one tetramer interacts with the - interface of an adjacent tetramer to create an interaction surface of 2034 Å². Much of this interface is established by four otherwise disordered linkers that adopt ordered conformations exclusively at the inter-tetramer interface (Supplementary Fig. 11). Loop 1, which is comprised of Ala1556-Asp1587 that connect ARM2 to the CLD, and loop 2, which is comprised of Pro1003-Met1023 in the CLD, contribute to the - side of the interaction (Fig. 9c). Loop 3 (Phe1036-Met1044) and loop 4 (Ser679-Glu690), both of which are in the CLD, contribute to the opposing + side of the interaction.

Higher-order assemblies were notably absent from the other states. By examining potential inter-tetramer interfaces, we found that the ability to form both the + and - interactions was unique to the inhibited state (Fig. 9d–e). The conformational restrictions imposed by an intact BTF ring in the preactivated, preactivated+Ca²⁺, and activated states permit each tetramer to form only a single interaction with an adjacent tetramer while the extended position of ARM2 precludes

either interaction from occurring in the resting states. Although there is a substantial entropic cost to these linkers adopting stable conformations, their extensive interactions suggest that the enthalpic gains from their ordering result in an overall reduction of free energy. In the inhibited state the increased flexibility of the CD following BTF ring disruption may offset this entropic penalty.

Outside of the interfaces and BTF1, which was too poorly-ordered to model, the tetramers in the assemblies are similar to the isolated inhibited state with an all-atom RMSD of 1.2 Å. The largest differences are in the flexible cytosolic domains, which rotate slightly compared to their positions in the isolated inhibited state to enable both the + and - interfaces to interact with the neighboring tetramer. Despite the global structural similarity, the Ca²⁺-dependence of the assembled inhibited state differs from that of the isolated inhibited state with the assembled inhibited state reaching a plateau at 100 nM Ca²⁺ while the isolated inhibited state increases in abundance to 10 μM (Fig. 2). The alternative Ca²⁺-dependence of the assembled inhibited state suggests that it may be functionally distinct, perhaps serving as an alternative mechanism for channel inhibition, which may rationalize how Ca²⁺ oscillations, which require both activation and high-Ca²⁺ feedback inhibition[48–51], can be detected in cells expressing the CD site mutant (Fig. 4e–g). Consistent with the assembled inhibited state potentially being functionally distinct from the isolated inhibited state, we did not observe densities that could be attributed to Ca²⁺ ions in the CD sites although we did observe densities corresponding to ATP, IP₃ and the JD Ca²⁺ ion.

## Flexibility of the C-terminal domain is driven by sampling acidic patches on the BTF ring

The CTD forms a four helix coiled-coil that extends through the center of the CD, connecting the JD to the BTF ring in its intact conformations (Supplementary Fig. 18 and Supplementary Table 9). While functional analyses of the CTD have provided conflicting results[65,69,70], its central

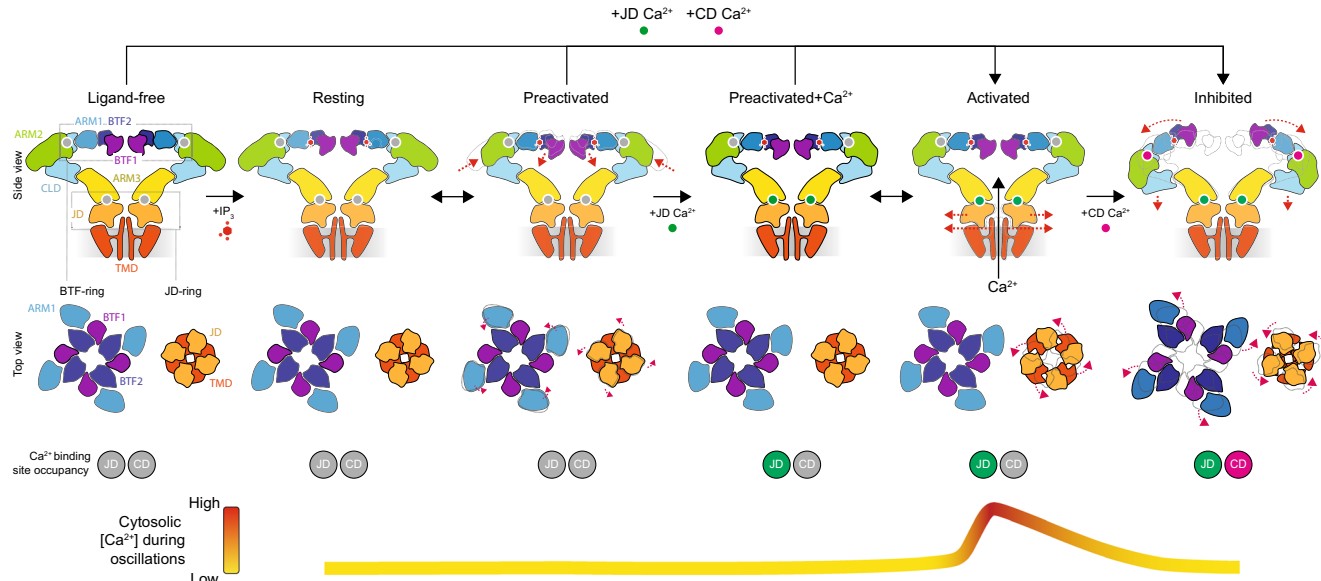

**Fig. 10 | Model for biphasic regulation of IP₃Rs by cytosolic Ca²⁺.** Schematic representations depicting the mechanisms of Ca²⁺- and IP₃-dependent activation and Ca²⁺-dependent inhibition of IP₃Rs. Row 1: Side views of the major states with front and rear protomers removed for clarity. Row 2: Cytosolic views of BTF-ring (left) and JD-ring (right). Magenta arrows highlight movements compared to previous state. Row 3: Occupancy of Ca²⁺-binding sites. Row 4: Correspondence between conformational state and cytosolic Ca²⁺ during Ca²⁺ oscillations.

position led to the proposal that it may serve as an allosteric link between the IP₃-binding sites in the CD and the pore[30]. In hIP₃R3, the CTD is poorly resolved due to its flexibility. Focused refinement and 3DVA revealed that a portion of the CTD of hIP₃R3 alternatively interacts with eight negatively charged patches on the inside of the BTF ring (Supplementary Fig. 18). While the limited resolution precludes building a model for the CTD, a conserved region of positively charged residues from Arg2654 to Arg2659 is the most likely candidate to bind to the negative patches on the BTF ring. The CTD adopts two conformations which are most apparent in the activated state, interacting with four of the eight patches in either ~C2 or ~C4 configurations (Supplementary Fig. 18 and Supplementary Movies 1-8), a noteworthy coincidence given the ~C2 and ~C4 TMD transition states. We investigated the essentiality of the CTD by truncating the channel at Leu2629 and monitoring the effects on IP₃R-mediated Ca²⁺ oscillations. We found that while the CTD deletion (dCTD) mutant produced Ca²⁺ oscillations with a rising-phase slope that is comparable to wild-type channels, the mean inter-spike interval of 12.2 seconds is significantly shorter (Supplementary Fig. 14). Therefore, while the CTD is not essential for channel activity, CTD deletion does alter Ca²⁺ dynamics in cells.

## Discussion

Here we defined the conformational landscape that underlies the biphasic Ca²⁺-dependence of IP₃Rs and gives rise to IP₃R-dependent Ca²⁺ oscillations in cells. Ordering the states based on their Ca²⁺ dependence frames a model for the ligand-dependent activation and inhibition of IP₃Rs (Fig. 10). IP₃ generated in response to extracellular stimuli can bind to the ligand-free channel without altering its global conformation, yielding the low-energy resting state. Once bound to the resting state, IP₃ enables the progression through the resting-to-preactivated transitions to the higher energy preactivated state, which appears to have a greater affinity for Ca²⁺. The increased affinity of the preactivated state for Ca²⁺ would promote binding of the ~100 nM basal cytosolic Ca²⁺ at the JD site, unlocking the JD ring and favoring the transition through the ensemble of high-energy intermediate states along the trajectory to the fully-open activated state. Upon opening, IP₃Rs release Ca²⁺ in the cytosol where it can bind to the low-affinity CD

site and stabilize the inhibited state to terminate Ca²⁺ release. With IP₃Rs closed, SERCA would be able to pump Ca²⁺ back into the ER and restore basal Ca²⁺ concentrations. As Ca²⁺ is sequestered back into the ER, Ca²⁺ can dissociate from the low-affinity CD site. When the BTF ring reforms, subsequent Ca²⁺ release events can then be initiated if IP₃ remains abundant, resulting in regenerative Ca²⁺ oscillations.

Thus, the conformational landscape of hIP₃R3 is comprised of multiple structurally distinct closed states and seemingly only one open state. Notably, ligand binding is not sufficient to determine conformational state, as distinct states exhibit identical ligand-binding profiles. For example, the preactivated+Ca²⁺ and activated states both bind IP₃, ATP, and Ca²⁺ at the JD site, yet the pore is closed in the preactivated+Ca²⁺ state and open in the activated state (Fig. 1). Similarly, the resting and preactivated states, as well as the intermediate resting-to-preactivated transitions, all bind IP₃ and ATP, but not Ca²⁺ (Fig. 1). Thus, the free energy gains associated with ligand binding are insufficient to drive the ligand-induced conformational changes, such as priming and gating, to completion. Rather, ligand binding biases the conformational equilibrium to increase the favorability of the high-energy states along the trajectory to activation.

While the trajectory to activation is populated with numerous high-energy states, the resting and inhibited states serve as the lowest energy states in the low and high Ca²⁺ conditions, respectively. As both of these states are closed, the resting and inhibited states contribute significantly towards establishing the biphasic Ca²⁺ dependence of IP₃Rs. This energetic landscape presents a highly-tunable system where post-translational modifications, protein-protein interactions, cellular metabolites, and other forms of regulation may shift the balance of states to modulate activation or alter the frequency and amplitude of Ca²⁺ waves without disturbing the principal biphasic Ca²⁺ dependence of the channel[19,21,55,71–73]. Consistent with this model, we identified several mutations that change the conformational landscape resulting in altered IP₃R-dependent Ca²⁺ oscillation dynamics without abolishing activation or inhibition of the channel (Figs. 4e–g, 5j–l, 7g–i and Supplementary Fig. 12). We also identified a second set of mutations that abolished Ca²⁺ oscillations, likely by removing one or more critical states from the conformational network (Figs. 4h–j, 8d–f and Supplementary Fig. 12). Thus, our structural landscape provides a

framework for understanding how diverse stimuli modulate Ca$^{2+}$ dynamics in cells[24,55,74].

Electrophysiological analyses have demonstrated that inhibition of IP$_3$R3 is highly cooperative, while activation is not[19,75]. These observations are consistent with the existence of multiple asymmetric states along the trajectory to activation and the complete absence of states along the trajectory to inhibition. The absence of any channels with partially disrupted BTF rings likely arises from the strain that accompanies Ca$^{2+}$ binding. Once even a single interface in the BTF ring is disrupted, the strain throughout the channel may cause the other interfaces to be pulled apart, resulting in the inhibited state. While the CD adopts several asymmetric states in the resting-to-preactivated transitions, all of the observed states display at least two-fold pseudosymmetry in the TMD. The lack of asymmetric TMD conformations may arise from the domain-swapped arrangement of the S1-S4 domain with respect to the pore, which assures cross-protomer communication. Similar ~C2 states have been observed for TRP channels, which share the domain-swapped 6TM fold[76-78].

Structural titrations have been previously performed using two ligand-gated ion channels, Slo2.2 and GIRK[47,79]. Intermediate states were noticeably absent from these analyses, which led to the conclusion that the gating processes were highly cooperative[47,79]. As electrophysiological analyses of Slo2.2 and GIRK also demonstrate this cooperativity, the correspondence between the structural and functional titrations of Slo2.2, GIRK, and IP$_3$R ion channels indicates that structural titrations can provide mechanistic insights into the processes that underlie protein function.

It has long been appreciated that IP$_3$Rs in cells can function in clusters of multiple channels that display a high degree of synchronicity[80]. These clusters have been shown to arise in an IP$_3$- and Ca$^{2+}$-dependent fashion[81,82]. While our structural interpretation is largely consistent with the functional and cellular evidence, it is unclear if the higher-order assemblies of inhibited channels that we observe correspond to the clusters that have been observed in cells. As the Ca$^{2+}$-dependency of the inhibited particles in higher-order assemblies differs from those of isolated inhibited state particles, it will be interesting to investigate if the formation of higher-order assemblies contributes to Ca$^{2+}$-dependent inhibition in cells.

Altogether, our analyses show how structural titrations, the process of determining structures in the presence of varying concentrations of regulatory ligands and co-factors, can reveal how stimuli bias the conformational landscape to modulate protein function.

## Methods

### hIP$_3$R3 expression

All constructs were N-terminally tagged with 10xHis followed by EGFP (Ca$^{2+}$ imaging) or mVenus (cryo-EM)[83] followed by a human rhinovirus 3C protease[84] cut-site and then human type 3 IP$_3$R. Plasmids were transformed into DH10Bac cells to generate bacmids[29]. 100–200 µg of purified bacmid were incubated with 400 µg of 25,000 MW polyethyleneimine (PEI; Polysciences Cat# 23966) in 1.4 mL water at 55 °C for 30 minutes to sterilize, then added to 50 mL of Sf9 cells at 1×10$^6$ cells/mL grown in suspension at 27–30 °C. The Sf9 TNM-FH (Grace's modified) media was supplemented with 1% penicillin/streptomycin, 0.1% Pluronic F-68 nonionic surfactant (Gibco Cat# 24040), and 4-8% fetal bovine serum to stabilize the virus. Virus titer was amplified to P3 and separated from cell debris by centrifugation. P3 virus was used to infect mammalian HEK293S GnTI$^-$ (ATCC CRL-3022) cells at a density of 3×10$^6$ cells/mL at a ratio of 50 mL virus for 800 mL cells and simultaneously stimulated with 4.5 mM valproic acid (VPA; Sigma Cat# P4543). Pellets were harvested from cells by centrifugation at 48–72 hours after infection and snap frozen.

### hIP$_3$R3 purification

All surfaces, vessels, and transfer plastics were washed extensively with reverse osmosis water prior to use to minimize contaminating Ca$^{2+}$. Membrane proteins were solubilized from 2.4 L of pelleted HEK293S GnTI$^-$ cells expressing wild-type hIP$_3$R3 for 2 hours by rotation in 2% lauryl maltose neopentyl glycol (LMNG; Anatrace Cat# NG310), 150 mM sodium chloride (NaCl), 20 mM HEPES pH 7.5, 1 mM phenylmethylsulfonyl fluoride (PMSF), 2.5 µg/mL aprotinin (Sigma Cat# A1153), 2.5 µg/mL leupeptin (Alfa Aesar Cat# J61188), 10 µg/mL pepstatin A (GoldBio Cat# P-020-25), 0.5 mM 4-benzenesulfonyl fluoride hydrochloride (AEBSF; EMD Millipore Cat# 101500), and a few flakes of lyophilized deoxyribonuclease (DNAse; Worthington Biochemical Cat# LS002139). The resulting cell lysate was centrifuged at 75,000x $g$ for 40 minutes. The supernatant was incubated with sepharose-coupled GFP nanobody affinity purification beads for 4 hours with gentle agitation[85]. The protein-GFP-nanobody-bead mixture was isolated in a column and washed with 50 mL of gel filtration buffer containing 150 mM NaCl, 50 mM Tris-HCl pH 8.0, 0.02% LMNG, and 2 mM dithiothreitol (DTT). The protein was eluted from the affinity column by cleavage with genetically modified human rhinovirus 3C protease overnight. Size exclusion chromatography was performed with a Superose 6 Increase column and the resulting protein peak was pooled and concentrated to 20 mg/mL in a 1 mL, 100 kDa MWCO concentrator (Cytiva VivaSpin Cat# 28932258).

### Structural titration sample preparation

Cryo-EM sample blotting paper contributes a significant quantity of contaminating Ca$^{2+}$ to protein preparations. We opted to produce our own low-Ca$^{2+}$ blotting paper by treating standard blotting paper (Ted Pella Standard VitroBot Blotting Paper Cat# 47000-100) with an extensive washing protocol. Over several days and multiple buffer exchanges, we treated with approximately 6 L of 100 µM EGTA in reverse osmosis (RO) water, then 6 L of RO water with Ca$^{2+}$ chelating beads (BIO-RAD Chelex 100 Resin Cat#142-1253), and finally 6 L of RO water alone. The treated paper was then stacked between extensively washed glass plates and subjected to vacuum for 24 hours to remove moisture and resume a flat shape. The treated filter paper is predicted to contain substantially less than 1 mM contaminating Ca$^{2+}$ (predicted starting condition of blotting paper[29]) and 100 µM residual EGTA (first wash condition).

To further control our sample Ca$^{2+}$ concentrations, we engineered a 5X ligand and Ca$^{2+}$ chelator cocktail. By combining 2 mM each of EDTA (K$_d$ 30 nM), EGTA (K$_d$ 127 nM), BAPTA (K$_d$ 153 nM), HEDTA (K$_d$ 4.8 µM) with 1 mM of ATP (K$_d$ 183 µM), we calculate that our buffer ensures a semi-log-linear relationship between free and total Ca$^{2+}$ from 1 nM to 300 µM[86]. The least well-controlled range for free Ca$^{2+}$ was between 1 nM and 10 nM requiring addition of 864 µM total Ca$^{2+}$, and the largest was between 10 µM and 100 µM, requiring addition of 2.0 mM total Ca$^{2+}$. Thus, our total contaminating Ca$^{2+}$ must be greater than 864 µM to generate a maximum 1-log-fold error in our target free Ca$^{2+}$ across the entire titratable range, ensuring that we maintain the semi-log-linear relationship between free and total Ca$^{2+}$ despite contaminating Ca$^{2+}$. To minimize the impact of widely varying kinetic properties of the chelators, we premixed the 5X solution containing 10 mM of each chelator, 5 mM ATP, 1 mM IP$_3$, and 2.5 mM fluorinated fos-choline-8 (Anatrace Cat# F300F), a detergent that does not interact with hydrocarbons, to protect the protein from the air-water interface. Sensitivity analysis using MaxChelator (https://somapp. ucdmc.ucdavis.edu/pharmacology/bers/maxchelator/webmaxc/ webmaxcE.htm) revealed that inaccurate pH was the largest contributor to deviations from the predicted free Ca$^{2+}$, and thus we carefully adjusted all solutions to pH 8, and added an additional 50 mM Tris pH 8.0 to the ligand and Ca$^{2+}$ chelator cocktail. CaCl$_2$ and MgCl$_2$ were added in varying quantities to generate the desired free Ca$^{2+}$ concentration and a constant 3 mM free Mg$^{2+}$ concentration. During grid

preparation, 3.2 μL of purified protein was added to the grid and incubated for 30 seconds, after which we added 0.8 μL of the ligand and $Ca^{2+}$ chelator cocktail directly to the droplet on the grid, immediately blotted with our low-$Ca^{2+}$ blotting paper for 2 seconds, then plunge-frozen using a ThermoFisher Vitrobot Mark IV. Since the $Ca^{2+}$ and chelators are premixed, the free $Ca^{2+}$ is at equilibrium in the master mix, and pipetting error when adding to the protein on the grid will have no effect on free $Ca^{2+}$. The only deviations due to pipetting error would be [IP$_3$] and [ATP], both of which are above saturating concentrations and so we assume those to be inconsequential for this analysis. Due to the non-equilibrium nature of grid preparation (e.g. evaporation, temperature changes) we acknowledge that while the semi-log-linear relationship established across the titration range is very precise, there is likely inaccuracy in our nominal free $Ca^{2+}$ concentrations. The final grid conditions have varying free $Ca^{2+}$, but constant 200 μM IP$_3$, 1 mM ATP, 3 mM free $Mg^{2+}$, 1.6 mM dithiothreitol (DTT), 2 mM EDTA, 2 mM EGTA, 2 mM BAPTA, 2 mM HEDTA, 50 mM Tris pH 8.0, 120 mM NaCl, 500 μM fluorinated fos-choline-8, and 159 μM LMNG.

### Fura-2 calibration for $Ca^{2+}$ chelator plus ligand cocktail

$Ca^{2+}$ calibration curves were generated using Fura-2 (Thermo Scientific Chemicals; J63686.MCR) and Invitrogen Calcium Calibration Buffer Kit (#C3008MP). Briefly, we prepared 50 μL samples for multiple specific free $Ca^{2+}$ concentrations by mixing buffer A (50 μM Fura-2, 10 mM EGTA, 100 mM KCl, 30 mM MOPS, pH 7.2, 3 mM $MgCl_2$) and buffer B (50 uM Fura-2, 10 mM Ca-EGTA, 100 mM KCl, 30 mM MOPS, pH 7.2, 3 mM $MgCl_2$) in the manner prescribed by the manufacturer. Fura-2 fluorescence emission at 510 nm was collected for an excitation scan from 280 to 450 nm in a clear bottom 96-well plate (Greiner bio-one; #655076) using Molecular Devices SpectraMax M5e microplate reader at room temperature. A calibration curve for $Ca^{2+}$ was generated by calculating the fluorescence emission ratios at 340 and 380 nm (Supplementary Fig. 1). Following this, 10 μL of the 5X ligand and $Ca^{2+}$ chelator cocktail (described earlier) with a nominal free $Ca^{2+}$ concentration of $10^0$-$10^4$ nM was added to 40 μL of 62.5 μm Fura-2 (ThermoFisher; #F-1200) diluted in gel filtration buffer (150 mM NaCl, 50 mM Tris-HCl pH 8.0, 0.02% LMNG, and 2 mM dithiothreitol (DTT)) to achieve a final concentration of 50 μM Fura-2 and 1X $Ca^{2+}$ chelator cocktail. Samples were excited at 340 nm and 380 nm and fluorescence emissions were collected at 510 nm using a Molecular Devices SpectraMax M5e microplate reader at room temperature. GraphPad Prism 9 was used to estimate free $Ca^{2+}$ concentrations by interpolating the fluorescent emission ratios at 340 nm and 380 nm excitations (Supplementary Fig. 1). Data presented are from three technical replicates.

### Cryo-EM data collection, analysis and model building

Images were collected at 0.826 Å/px magnification on an FEI Krios with Gatan K3 detector at 15 e$^-$/pix/sec with 3 sec exposure (0.05 sec/frame) for a total dose of 66 e$^-$/Å$^2$ in automated fashion using SerialEM[87,88]. Five datasets were collected during the same session for each $Ca^{2+}$ concentration on a series of grids that were prepared sequentially resulting in 637 movies at 1 nM, 2150 movies at 10 nM, 6126 movies at 100 nM, 1372 movies at 1 μM, and 3136 movies at 10 μM. A sixth dataset of 4312 movies collected at nominal 100 nM free $Ca^{2+}$ from a grid prepared later in the sequence was collected as a technical replicate to assess experimental error (Supplementary Fig. 18).

All movies were combined and processed starting in CryoSparc Live v3.3.1 for motion correction, CTF estimation, and bias-free autopicking at a rate of 380 picks/micrograph with a gaussian blob of dimensions between 166 and 240 Å, corresponding to the smallest and largest diameter of the known conformational states of IP$_3$Rs. Thus, all of the following classification decisions were made in aggregate and without any a priori knowledge of the dataset from which

particle subsets were derived. The over-picked particle stack was extracted in a 512 box and subjected to iterative CryoSparc v3.3.1 Heterogeneous Refinement[89] without imposing symmetry. Four references corresponding to the resting, activated, inhibited, and a single consensus average of the preactivated +/- $Ca^{2+}$ states were used as inputs. These references were previously determined from the combined data using traditional single-particle approaches. The remaining eight input classes were pure noise decoy references generated by randomly sampling a very small number of particles via CryoSparc v3.3.1 Ab-Initio without alignment. The decoy references attract false positives, while the four high-resolution references attract true positives. These references were used for all classifications described herein.

After several rounds of "decoy" classification without imposing symmetry, the particle stack went from 7.8 M particles to 1.7 M particles, with 351k, 117k, 145k, and 1045k residing in the classes obtained from the resting, preactivated, activated, and inhibited references respectively. 2D classification of the discarded classes confirmed that no unintentional removal of true positives occurred. At this stage, each stack was independently subjected to an additional iteration of classification to allow fine separation of states whereby the non-self-references attract particles away from the self-identifying class in cases where the particles deviate from the consensus state in subtle ways. This resulted in six classes that are depicted in the second tier of the cryo-EM workflow figure (Supplementary Fig. 2), with classes that refined to worse than 7 Å being discarded as junk or damaged particles.

Each of these six stacks were refined enforcing C4 symmetry to improve signal for reference-based corrections prior to Bayesian Polishing in Relion v3.1.3[90]. At this stage, optical groups were separated and both local and global CTF parameters were optimized in CryoSparc v3.1.1 during Non-Uniform Refinement[91] procedures. Due to the very large number of optical groups, it was found that the fourth-order terms of spherical aberration and tetrafoil[92,93] were not being fit accurately in some groups, and hence we did not fit these terms. In aggregate the per particle, per micrograph, and per optical group corrections resulted in improvements for the resting-like stack with strong TMD density (231k particles; 3.5 Å to 2.7 Å), resting-like stack with weak TMD density (108k particles; 3.9 Å to 3.3 Å), preactivated-like stack with weak CD density (83k particles; 4.0 Å to 3.6 Å), activated-like stack (65k particles; 3.7 Å to 3.1 Å), preactivated-like stack with weak TMD density (76k particles; 3.8 Å to 3.2 Å), and inhibited-like stack (1045k particles; 3.2 Å to 2.5 Å). These stacks were subjected to one final round of classification revealing the five primary C4 symmetric states called resting (192k particles; 2.8 Å), preactivated (47k particles; 3.7 Å), preactivated+$Ca^{2+}$ (31k particles; 3.6 Å), activated (56k particles; 3.1 Å), and inhibited (917k particles; 2.5 Å) states and several heterogeneous conformational ensembles. Notably, local conformational changes can still be observed in the five major states by 3DVA at the end of the hierarchical classification due to the large size and overall flexibility of the channel. Our final maps and the corresponding models represent the average of the particles that comprise these states.

We further improved the C4 symmetric states by performing C4 symmetry expansion and local refinement to correct for subtle local asymmetries in the particles. We used a model to precisely delineate masks surrounding modular units that flex and move in unison: (1) a mask containing a single chain from the tetramer (2) the entire cytosolic domain consisting of residues 1-1697 from a single chain (3) BTF1, BTF2, and ARM1 consisting of residues 1-664 from a single chain (4) CLD, ARM3 consisting of residues 665-1100 and 1586-2074 from a single chain (5) ARM2 consisting of residues 1101-1586 from a single chain (6) TMD, JD consisting of residues 2111-2611 from a single chain. The masks generated from these models were dilated by 4 pixels and a cosine soft-edge was applied for 40 pixels, thereby avoiding ringing and mask artifacts that occur when converting hard

edges in real-space to reciprocal space. Therefore, this mask retains 100% of the information at ~3 Å away from the model, and 50% of the information at ~25 Å away from the model. CryoSparc v3.3.1 Local Refinement resulted in resolutions ranging from 2.5 Å (TMD/JD) to 3.3 Å (ARM2) for the resting state, 3.6 Å (TMD/JD) to 6.5 Å (ARM2) for the preactivated state, 3.3 Å (BTF1/BTF2/ARM1) to 4.2 Å (ARM2) for the preactivated+$Ca^{2+}$ state, 2.9 Å (BTF1/BTF2/ARM1) to 3.3 Å (ARM2) for the activated state, and 2.5 Å (TMD/JD) to 3.4 Å (BTF1/BTF2/RM1) for the inhibited state (Supplementary Figs. 3, 5, 6, 7 and 9). In some highly-heterogeneous cases, the local refinements were subjected to a procedure that will be described in the treatment of the conformational ensembles to improve the resolution (e.g. BTF1/BTF2/ARM1 in the inhibited state).

The local refinements were independently subjected to Phenix v1.20.1-4487 Resolve Cryo-EM[94] guided only by experimental density (no model) and employing a lenient mask that contains all proteinaceous and detergent micelle density, an approach we have used previously[95–97]. As part of the procedure, the final maps are sharpened using a half-map derived factor. The resulting density modified and sharpened maps were cropped to a single chain and used for iterative model building using coot[98], ISOLDE[99] and composite map generation using a 20-residue sliding window cross-correlation (Phenix v1.20.1-4487 Combine Focused Maps)[100], which we found to produce artifact-free maps when compared to Chimera's 'vop maximum' command[101]. For the highest resolution composites (resting, activated, and inhibited) the density-modified local refinements were super-sampled prior to composite generation to aid interpretation of ligands, ions, and waters. Inspection of the resulting composite maps showed that they were free of model-based overfitting, for example density for ions, ligands, and lipids remain intact despite being removed from the input model. The final models were refined against the composite map with Phenix v1.20.1-4487 Real-Space Refinement[102].

The remaining classes represent highly-heterogeneous conformational ensembles that we interrogated via 3D variability analysis (3DVA)[36]. We relaxed our assumptions about symmetry by performing C4 symmetry expansion on each class. For the resting-like ensemble with weak CD density, resting-like ensemble with weak TMD density, preactivated-like ensemble with weak CD density, and the preactivated-like ensemble with weak TMD density, we performed 3DVA with a full channel mask and filter resolution between 5 and 8 Å and clustered each of 3 modes independently into 5 groups. Occasionally, one or two clusters would be populated with very few particles, suggesting that a fewer number of clusters was adequate to represent the underlying heterogeneity. We then refined each class (CryoSparc v3.3.1 Local Refinement due to symmetry expansion) and assessed the resulting structures, selecting the mode of variability that contained our features of interest. From the resting-like and preactivated-like stacks with weak TMD density, we obtained the ~C2 and ~C4 TMD transition states presented in Fig. 7d–e and Supplementary Fig. 8. From the resting-like and preactivated-like stacks with weak CD density, we obtained the asymmetric ARM2 sampling states presented in Fig. 5a. For the ARM2 retraction analysis in Fig. 5b–g and wedge loop analysis in Supplementary Fig. 15, we increased the requested number of clusters to 10 and 20 respectively and selected 6 refinements that appeared to be on a shared trajectory for both ARM2 retraction and loop melting for presentation.

To calculate an octameric reconstruction of two adjacent tetramers in a higher-order assembly, the nonsymmetry expanded stack of 1045k inhibited state particles was subjected to CryoSparc v3.3.1 Heterogeneous Refinement seeded with 24 identical references of the inhibited state. Strong density for an adjacent tetramer was present in three of the resulting classes, totaling 246k particles. This classification was used to quantify the particle distributions for clustered versus isolated inhibited states in Fig. 2. The clustered particles were subjected to C4 symmetry expansion and local refinement using a mask encompassing the entire CD of one protomer, which suggested that the interaction was formed between ARM2 of the central protomer and CLD of the adjacent protomer. To separate the protomers that participate in the inter-tetramer interface from those facing away, we next performed 3DVA with two different masks on the symmetry expanded stack of 984k particles. We first generated a mask comprising the CLD of the central tetramer interacting with ARM2 of the adjacent tetramer to identify protomers that form what we call the + half of the interaction, which yielded 85k particles at 3.3 Å. Then, we generated a mask comprising ARM2 of the central tetramer interacting with the CLD of the adjacent tetramer to classify protomers that form what we call the - half of the interaction, which yielded 88k particles at 3.3 Å. These two populations represent two halves of the ~C2 symmetric interface between two assembled tetramers.

In addition to the masks covering the interfaces, we used the same set of local refinements that we used for the five C4 symmetric states to produce a well-resolved composite map of each protomer in the two associated tetramers. For each population, we first produced a consensus tetrameric alignment, and then proceeded to subject each protomer of the tetramer to the six local refinement masks that were used to produce the C4 symmetric inhibited state (entire protomer, TMD/JD, ARM3/CLD, ARM2, BTF1/BTF2/ARM1, and entire CD). Each of these local refinements was then subjected to Phenix Resolve Cryo-EM, and finally to Phenix Combine Focused Maps to produce a tetrameric composite map of each half of the interaction. For the depiction in Fig. 9a–b, we fit these two tetrameric composite maps together and used Chimera 'vop maximum' to create a single volume of the entire assembly.

Once all the particles were assigned to a specific state or ensemble in CryoSparc, we determined the number of particles that originated from each of the $Ca^{2+}$ concentrations via unique identifiers in the micrograph names in order to calculate the relative abundance of each state at each condition. Because we collected images from two grids prepared at 100 nM $Ca^{2+}$, we combined the particles and report the weighted average in Fig. 2. We compare the two data sets in Supplementary Fig. 17.

For the depictions of the composite maps in Fig. 1, four copies of the single-chain composite were fit to the consensus C4 refinement and combined using the Chimera 'vop maximum' command[101]. For the depictions of ARM2 density in Fig. 5, unsharpened local CD refinements were shown. For depictions of the overall density or slices at the JD ring, gate, and S1'-S1" in Fig. 6 and Supplementary Fig. 8, the unsharpened consensus refinement maps were low-pass filtered to 4 Å (overall) or 7 Å (zoomed) using 'relion_image_handler'[103,104]. For the depictions of the wedge loop density in Supplementary Fig. 15, the resting state composite map was used. For the depictions of the wedge loop density in Supplementary Fig. 15 a B-factor derived from the Guinier plot was used to sharpen the CD local refinements for presentation. All figures depicting models were generated in PyMol (Schrodinger, LLC. 2010. The PyMOL Molecular Graphics System, Version 2.5.3), and all figures depicting density alone were generated in ChimeraX[105,106]. For the electrostatics calculations in Supplementary Figs. 16 and 18, the Adaptive Poisson-Boltzmann Solver (APBS) algorithm[107] was utilized via PyMol plug-in.

## High performance computing

The MSK High Performance Computing (HPC) resource provides a GPU cluster built for computing large volume data over a range of applications from drug discovery to deep learning and image processing. It contains 120 nodes connected by a 100 Gigabit ethernet backbone. The nodes used for this project each contain Intel Xeon Platinum 2.2 GHz CPUs and 1 TB DDR4 RAM. Each node also contains four A100 GPUs interconnected using NVLink. The cluster runs the CentOS operating system and is supported by a 4 PB high-speed GPFS-based parallel filesystem. A 200 TB NVMe-based Weka ultra-fast tier

was used as scratch space. The CPU to GPU communication is established over PCIE 4.0. The project used IBM Spectrum LSF as the orchestrator of shared resources and parallelization is further achieved by MPI over the ethernet network. All cryo-EM software excluding CryoSparc was maintained via HMS SBGrid[108]. Multiple sequence alignments were performed using the MUSCLE[109] algorithm in DNASTAR LaserGene MegAlign Pro 17.3.

## Cloning

Site-directed mutagenesis and cloning for all hIP₃R3 constructs detailed in this study was performed using either two concurrent single-primer reactions (PMID: 19566935) or NEBuilder HiFi DNA Assembly kit (NEB Cat# E2621) in Mach1 T1 cells (ThermoFisher; Cat#C862003). Oligonucleotides (Supplementary Table 12) were designed manually for single-primer reactions, whereas NEBuilder (https://nebuilder.neb.com/) was utilized for designing oligos for mutagenesis based on NEBuilder HiFi DNA Assembly. All oligos were acquired from Integrated DNA Technologies.

## Adherent cell culture

HEK293T IP₃R-null cells were obtained through Kerafast[52] and cultured to a confluency of ~75-80% on 100 ×20 mm tissue culture treated dishes in DMEM supplemented with 10% fetal bovine serum, 100 U/ml penicillin, 100 mg/ml streptomycin at 37 °C with 5% $CO_2$. For imaging, cells were then split in a 1:4 ratio and plated on poly-D-lysine coated, 35 mm diameter, optical quality glass-bottom culture dishes (World Precision Instruments; # FD35PDL-100) and incubated for ~18-24 hours. At ~60% confluency, cells were transduced with a 200 μl baculovirus followed by incubation at 37 C, 5% $CO_2$ for another 24 hours. All constructs used for $Ca^{2+}$ imaging in this study were overexpressed in HEK293T IP₃R-null cells using the BacMam system[110].

## Ca²⁺ imaging and data processing

24 hours after baculovirus transduction, cells were gently washed with imaging buffer [20 mM HEPES supplemented $Ca^{2+}$, $Mg^{2+}$ free, Hank's balanced salt solution (ThermoFisher; #14175103)] followed by incubation for 1 hour at 37˚C and 5% $CO_2$ in 1.8 ml of imaging buffer containing 3 mM Cal-520-AM (AAT Bioquest; #21130) Cal-520-AM-loaded cells were removed from the $CO_2$ incubator and equilibrated at room temperature for 5 minutes prior to IP₃ stimulation by the addition of 200 μl of 1 mM carbachol (Alfa Aesar; #L06674-06), a Gαq-coupled M3 muscarinic receptor agonist. Carbachol was added at least 10 mm away from the imaging site and allowed to diffuse to a final concentration of 100 μM. Movies of carbachol-induced $Ca^{2+}$ release in cells were collected at 20x with LD Plan-Neofluar 20X/0.4 Korr M27 objective, for 10 minutes, at 3×3 binning (912×736 pixels post binning), with an exposure time of 250 ms on a Zeiss Axio observer D1 inverted phase-contrast fluorescence microscope equipped with an Axiocam 506 Mono camera (Zeiss). Cal-520-AM imaging was carried out by exciting the sample at 493 nm and monitoring emission at 515 nm using X-Cite Series 120Q illumination system and Zeiss filter set 38 HE.

Ca²⁺ imaging movies were processed using ImageJ[111], Fiji[112] and MathWorks MATLAB 9.12.0.1884302 (R2022a) to extract Cal-520-AM fluorescence traces from individual cells. Movie stacks were background-subtracted with a 200-pixel rolling ball radius in ImageJ. Maximum intensity projection of the stack was used to generate a difference of gaussian image, which was used for edge detection and cell segmentation using MATLAB's Image Processing Toolbox. Traces were then extracted from segmented cells, smoothed over 41 frames using a Savitzky−Golay filter of polynomial order 2, normalized by Z-score, and baseline adjusted using the linear method of MATLAB's 1-D data interpolation function with a custom MATLAB script called Baseline Fit[113]. In the baseline-adjusted traces, the smallest observed $Ca^{2+}$ oscillation peak value was used to manually threshold and identify other peaks automatically. Detected peaks were then used to calculate inter-spike intervals using MATLAB's Signal Processing Toolbox. All statistical tests were performed using GraphPad Prism 9. Data reported are from 3 independent biological replicates.

For analysis of peaks from individual replicates, traces with transients/oscillations were baseline adjusted in MATLAB using Baseline Fit[113] and normalized between 0 and 1. The first peak of each oscillation/transient was identified and a window of 75 seconds on both sides of the peak was extracted and aligned at the peak position. Mean and 95% confidence intervals were calculated using GraphPad Prism and overlayed on traces from a single biological replicate. A 1 second window on both sides of the mean data point corresponding to half maximal intensity were fit to a straight line and used to calculate the mean rising phase for constructs exhibiting transients/oscillations. Traces with oscillations were sorted based on the maximum number of distinguishable peaks and plotted as a fraction of total oscillatory traces.

## Reporting summary

Further information on research design is available in the Nature Portfolio Reporting Summary linked to this article.

## Data availability

The Cryo-EM data and atomic coordinates generated in this study have been deposited in the Electron Microscopy Data Bank and Protein Data Bank under accession codes EMD-41323 and 8TK8 for the Resting state structure, EMD-41347 and 8TKD for the Preactivated state structure, EMD-41348 and 8TKE for the Preactivated+$Ca^{2+}$ state structure, EMD-41349 and 8TKF for the Activated state structure, EMD-41350 and 8TKG for the Inhibited state structure, EMD-41351 and 8TKH for the Labile resting state 1 structure, EMD-41352 and 8TKI for the Labile resting state 2 structure, EMD-41366 and 8TLA for the Higher-order inhibited state 1 structure and EMD-41365 and 8TL9 for the Higher-order inhibited state 2 structure. Cryo-EM data generated in this study have been deposited in the Electron Microscopy Data Bank under accession codes EMD-41324, EMD-41325 and EMD-41326 for the Resting-to-Preactivated transition states, EMD-41327, EMD-41328, EMD-41329, EMD-41330, EMD-41331 and EMD-41332 for the ARM2 retractions states, EMD-41333, EMD-41334, EMD-41335, EMD-41336, EMD-41337 and EMD-41338 for the Wedge loop progression states, EMD-41339 for the ~C2 Preactivated TMD Transition, EMD-41340 for the ~C4 Preactivated TMD Transition, EMD-41341, EMD-41342 and EMD-41343 for the Activated CTD states, EMD-41344 for the ~C2 Resting TMD Transition, and EMD-41345 for the ~C4 Resting TMD Transition. The atomic coordinates of previously published structures used in this study at available at the Protein Data Bank under accession codes 3JAV, 6MU2, 7LHF and 7LHE for rat Type 1 IP₃R, 6DQJ and 6UQK for human Type 3 IP₃R in a resting state, 6DQS, 6DQZ, and 6DR0 for human type 3 IP₃R resting-to-preactivated transition states, 6DQV, 7T3P, 7T3Q, and 7T3R for human type 3 IP₃R in preactivated states, 7T3T for human type 3 IP₃R in an activated state, 6DRC, 6DR2, 6DRA, and 7T3U for human type 3 IP₃R in inhibited states, 1N4K, 3T8S, and 3UJ0 for fragments containing the IP₃-binding domain of rat type 1 IP₃R, and 5TAP for caffeine- and ATP-bound rabbit RyR1. Source data are provided with this paper. Plasmids are available upon request. Source data are provided with this paper.

## Code availability

Custom MATLAB scripts used for fluorescence-based $Ca^{2+}$ imaging data analysis are available at https://doi.org/10.5281/zenodo.8411195.

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

## Acknowledgements

We thank Jason de la Cruz at the Memorial Sloan Kettering Cancer Center (MSKCC) Richard Rifkind Center for cryo-EM assistance with data collection and the MSKCC High-Performance Computing (HPC) group, in particular Neeraj Harikrishnan and Jamie Cheong, for assistance with data processing. We thank Ellen Zhong for discussions about conformational heterogeneity in cryo-EM data and Elizabeth Campbell, Seth Darst, Melinda Diver and Stephen B. Long for comments on the manuscript. This work was supported by NIH NCI Cancer Center Support grant P30 CA008748 (R.K.H.), NIGMS R01-GM13230704 (R.K.H.), NCI F31-CA243235 (N.P.), the Searle Scholars Program (R.K.H.) and the Josie Robertson Investigators Program (R.K.H.).

## Author contributions

N.P., V.S. and R.K.H. conceptualized the project and contributed to writing the manuscript. N.P. performed the bulk of cryo-EM analysis. V.S. performed the bulk of optical Ca2+ imaging analysis. N.P., V.S. and R.K.H. assisted each other on all experiments and analysis.

## Competing interests

The authors declare no competing interests.
