## [Peer Review File · Nature Communications]

Structural titration reveals Ca²⁺-dependent conformational landscape of the IP3 receptorREVIEWER COMMENTS

Reviewer #1 (Remarks to the Author):

In the present manuscript, authors have done cryo-EM structural and functional characterization of type 3 IP3R (hIP3R3) at varying calcium concentrations spanning the physiological range. Through this work, authors have deduced conformational landscape that enables IP3Rs to pivot from activation to inhibition to generate Ca²⁺ oscillations thereby providing structural basis for IP3R-generated Ca²⁺ oscillations. Briefly, 5 major structures for resting, preactivated, preactivated+calcium, activated and inhibited states of hIP3R3 have been identified. The 5 structures have been analyzed for their ligand binding (IP3/ATP/Calcium/Zinc) profiles and their functional and mechanistic implications in generating calcium oscillations. Intermediate states along the activation to inhibition trajectory have also been identified through advanced cryo-EM image processing methods. Section describing higher order inhibited states is intriguing.

Cryo-EM structure determination and analysis is comprehensive and is supported with mutation and other biochemistry experiments.

Comments:

1. At several places in the text, figure for the activated state (Figure 1E) is cited for the inhibited/closed state. Figure 1F corresponds to closed state. e.g.,

Lines 118-119: *“In the fifth major state and the pore is closed (Figure 1E; Tables S1 and S2)”*

Lines 194-196: *“We observed two distinct inhibited states –higher-order assemblies (Figures 1E, S4 and S2).”*

Figure 7 can also be cited for assembled inhibited state in lines 194-196.

2. Lines 163-166: Appropriate figure citations from Figure 1 can be provided.

3. At several places in the text Tables are referred as Table S1, S2 etc. While the actual Tables are labeled as Table 1, 2 etc. e.g.,

Line 69-71: *“merged these datasets and performed image processing in aggregate (Figure S2; Table S1).”*

“Using hierarchical classification, we resolved five four-fold symmetric major states for hIP3R3 at resolutions up to 2.5 Å (Table S1 and S2).”

It is not clear if above statements are referring to Tables 1 and 2.

4. Figure S10: Mismatch between Figure panels and their description in the figure legend. e.g., Apo vs resting state in figure panel (A) is incorrectly described as resting and preactivated states in Figure text. Rest of the descriptions also do not match the Figure panels.

5. Figure 7H description mentions *“modeling of a hypothetical higher-order interaction of the preactivated, preactivated+Ca²⁺, or activated state”* but only activated state is shown in Figure 7H

6. High resolution for Table 8 can be provided to make text easily readable.

7. Lines 482-484: *“Consistent with BTF ring disruption being a key aspect of inhibition, mutations at the interface between BTF1 and BTF2 of the neighboring protomer can diminish or eliminate carbochol-induced Ca²⁺ oscillations in cells (Figure S12X).”*

Residues mutated at the interface between BTF1 and BTF2 can be mentioned.

Figure S12X refers to the effect of CTD deletion mutant. Figure for the effect of mutations at the BTF1 and BTF2 interface should be provided.

8. Lines 580-584: Suitable references supporting this statement can be provided.

9. Like Figure 3Q, a figure describing CD calcium binding site can be provided and cited for Lines 168-170.

10. Table 4 description mentions *“Validation of cryo-EM density presented in Figure 3 and Figure S13”*

However, Figure 3 has no cryo-EM density described in it and Figure S13 cryo-EM density validation is not presented in Table 4. In my opinion Table 4 describes validation of cryo-EM density mentioned in Figure 4

11. Table 5 description mentions *“Validation of cryo-EM density presented in Figure 4 and Figure S15”*

Are the authors intending to refer to Figure 6 instead of Figure 4. Figure 4 density does not match with the description provided in Table 5.

12. Table 6 description mentions *“Validation of cryo-EM density presented in Figure S17”*

Table 6 instead describes validation for density presented in Figure S13

13. Tables 4-6 are misleading and should be reworked.

14. Validation of cryo-EM density presented in Figure S17 is not provided.

15. In accordance with the Journal policy, *proper EMDB and PDB codes* should be provided in the Data and Code availability section and referenced in the manuscript and Tables.

16. Lines 321-327: Like other places in the text, terms *“rotation”* and *“tilt”* can be quantified here.

17. Figures 4 and 7: Resolution for the cryo-EM density maps presented in these figures can be mentioned. Mention if low pass filtered to enable comparison/description.

18. Figures 7G and S4F: Distinct hIP3R3 domains surrounding Loops 1-4 in higher order clusters can be labeled/colored to enable clear understanding of their positioning.

19. Lines 777-779: *“Inspection of the resulting composite maps showed that they were free of model-based overfitting, for example density for ions, ligands, and lipids remain intact despite being removed from the input model.”*

What is the nature, binding modes, presence, or absence states of lipids in the five major states of hIP3R3 reported namely, resting, preactivated, preactivated+calcium, activated and inhibited. Mention if any correlation between lipid binding and calcium concentrations/titrations in gating mechanism across these five states?

20. Lines 591-594: *“Electrophysiological analyses have demonstrated that inhibition of IP3R3 is highly cooperative, while activation is not 19,79. These observations are consistent with the existence of multiple asymmetric states along the trajectory to activation and the complete absence of states along the trajectory to inhibition.”*

Lines 602-604: *“The presence of multiple transition states resolved in our analysis of hIP3R3 thus contrasts with two prior structural titrations of the Slo2.2 and GIRK K⁺ ion channels where intermediate states were noticeably absent and the transitions from open to closed were highly cooperative processes 46,83”*

The above two statements do not bring out the contrast as mentioned in Lines 602-604 and imply that lack of multiple states makes inhibition a highly cooperative process across hIP3R3, Slo2.2, and GIRK.

21. Lines 180-182: *“Single-particle cryo-EM analysis of vitrified samples represents a near equilibrium assessment of their conformational landscape, allowing one to infer relative conformational free energy from the number of particles that populate specific structural classes.”*

How good is using particle counts as proxy for free energy at estimating conformational free energy landscapes compared to quantitative estimation methods?

22. Table 3. In the validation of higher order inhibited assembly, model composition mentions zero ATP/IP3 ligands in the higher order states. Are these ligands not observed or not modeled into the maps? It should be clarified in the text or table. Also, can be clarified is the reason to build only a fraction of hIP3R3 (2 chains, 11,906 atoms) in the maps for higher order inhibited states vs the isolated inhibited states (4 chains, 72,296 atoms).

23. Lines 203-206: *“Although the structures of the tetramers in the higher-order assemblies are indistinguishable from the individual inhibited tetramers, their divergent Ca²⁺-dependence suggests*

that they are distinct states and that formation of higher-order assemblies may represent an alternative mechanism for achieving an inhibited state, as we will discuss later.”

Lines 503-507: “Although the tetramers in these assemblies are globally quite similar to the isolated inhibited channels and densities can be observed in both Ca²⁺-binding sites, they display an alternative Ca²⁺ dependence, suggesting that channels in higher-order assemblies may be functionally and structurally distinct from isolated inhibited channels.”

If the complete models are not built into the higher order assemblies (comment 22), are the structural comparisons made only on density maps to state “structures of the tetramers in the higher-order assemblies are indistinguishable from the individual inhibited tetramers”?

If the structures of tetramers in higher order assemblies are indistinguishable from the isolated inhibited tetramers as stated in lines 203-206, what distinct structural features of tetramers between higher order and isolated states are being mentioned in lines 503-507?

24. Lines 144-145: “IP3 is coordinated by the same residues in both binding modes.”

It is interesting to see that despite ARM1 tilting towards IP3 to contact ARM1-BTF2 interface, IP3 coordinating residues remains same.

25. Lines 55-59: “We therefore sought to establish high-resolution thermodynamic models of IP3R activation and inhibition using single-particle cryo-EM. By collecting images of human type 3 IP3R (hIP3R3) vitrified in a broad range of Ca²⁺ concentrations and treating particle abundance as a proxy for the relative free energy of each state, we evaluate how Ca²⁺ biases the conformational landscape of IP3Rs. These results establish the structural basis for IP3R-generated Ca²⁺ oscillations.” 

The above statements generalize the results of this manuscript to IP3 receptors by conducting structure-based calcium titrations on type hIP3R3 receptors. Would type IP3R1 and IP3R2 channels be expected to exhibit similar conformational landscape with five major states with similar calcium binding profiles as described here for hIP3R3 receptors?

26. Lines 171-173: “Taken together with our previous analyses of hIP3R3 in saturating Ca²⁺ 29, these data are consistent with the JD and CD sites being the primary Ca²⁺ binding sites in IP3Rs.”

Following study identifies 5 calcium binding sites (Figure 3) in type 1 IP3R (IP3R1) channels at inhibiting calcium concentrations.

Fan, G., Baker, M.R., Terry, L.E. et al. Conformational motions and ligand-binding underlying gating and regulation in IP3R channel. *Nat Commun* 13, 6942 (2022).

It would be interesting to see if the residues defining other three calcium binding sites (besides JD and CD) are conserved across IP3Rs. Structural comparison can shed light on differential calcium binding profile of inhibited states of IP3R3 (2 calcium binding sites) and IP3R1 (5 calcium binding sites)

27. Are the second and third major states namely preactivated and preactivated+calcium states observed for the first time in this manuscript or similar structures are available in the literature? Resting, activated and inhibited states of hIP3R3 reported here are mentioned to be observed in previous studies on IP3Rs.

Reviewer #2 (Remarks to the Author):

This manuscript describes the conformational landscape of the IP3 Receptor, a well-known calcium release channel that resides in the ER membrane. The authors used cryo-EM to perform titrations of the IP3R across a range of free calcium concentrations, carefully analyzing the distribution of particles in various distinct states. The authors nicely show that various states already exist at different concentrations, and that increasing levels of calcium change the relative distributions. They also provide very detailed descriptions of the structural transitions and show the importance of the different calcium binding sites using calcium imaging experiments.

Although many high-resolution cryo-EM studies of the IP3R have been described before, this is by far the most detailed and systematic study of IP3R structure and conformation, and even includes the discussion of asymmetric states. This is of high interest in the field, although I think that, as presented, the manuscript is also very dense in several sections, which may deter experts in the IP3R field who are not structural biologist themselves.

The study deserves a home in Nature Communications, but I have a series of suggestions for improvement.

Comments:

* The authors went to great care to avoid contaminating calcium from the blotting paper. However, it not clear what exactly has been done to experimentally verify the predicted free calcium concentration in the buffers. There is a short section “fura-2 calibration” in the methods, and Fig S1 shows fluorescence, but no experimentally determined concentrations are presented. The authors should list the measured free calcium concentrations in their buffers and include this in the Results. This could be checked with a calcium-sensing electrode, or with a calcium indicator, but using reference standards to obtain true calibration curves. If some of the calcium concentrations fall outside of the testable range, this should also be mentioned explicitly.

Note: Although there is biophysical interest in investigating conformations at 1nM free calcium, there is little physiological relevance as this is much lower than the resting calcium concentration in most cells. I recommend this rationale (biophysical purposes) to be mentioned up front. The authors could also point out resting calcium concentrations that IP3Rs would encounter in a native context, and indicate what this would mean for the expected conformational landscape in those conditions.

*The description of the calcium occupancy seems to be all or none, i.e. it was observed, or it was not observed. One can imagine that binding or not binding to a particular site correlates with the conformational state, but one would also expect asymmetry in at least some datasets that are not saturating, i.e. not all 4 equivalent binding sites in the tetramer are occupied simultaneously. Alternatively, when the degree of cooperativity in the four equivalent sites is extremely high, partial binding may never observed in any individual class. The authors have discussed this, mentioning that high cooperativity is known to exist for inhibition, but not for activation by calcium. Thus, one would expect some classes with less than four, but more than zero calcium ions bound at the activating binding site in some of the datasets. This could manifest as either lower apparent occupancy (when C4 symmetry is applied), or, potentially, as subtle asymmetric features in the subunits (for which data would have to be analyzed in C1).

I recommend the authors to describe the symmetry/asymmetry a bit better and further describe the possibility for partial occupancy of calcium. First, what criterion did the authors use to decide whether or not there is symmetry? Most classes are described as symmetric, and some classes with clear asymmetry were observed. Was any quantitation of asymmetry utilized prior to deciding whether it was C4?

For sites showing calcium density, what was the threshold level needed to observe the calcium? And what is the density relative to the average density (overall or local in the region around the calcium)?

Using this, are there signs of lowered calcium occupancy in some classes? If symmetry is not forced in the symmetric classes, are there indications for different calcium occupancies in the equivalent sites, possibly with additional subtle asymmetric features? This analysis may be outside of the limits of what could be dissected from the datasets, but I urge the authors to attempt this, or mention the results if this was already done with negative results, mentioning the limitations.

Note: Descriptions of the density thresholds used are also missing in most figures.

* readability: several sections are very dense, especially the descriptions on pages 13-16. This may deter many non-structural biologists in the IP3R field. Details matter, but the authors may want to provide more overall descriptive summaries in each section of the results, followed by the details (or move some details to supplementary and/or figure legends).

*The 'assembled inhibited state' is bizarre, and it is not clear if there is physiological relevance. In the ER membrane, there are 2D constraints imposed by the membrane. Figure 7E and 7F show that there would need to be an extreme curvature of the membrane that allows this, and this result may thus be an artefact from the preparation, where no membrane is present. Clustering of IP3Rs in ER membranes has been observed, but if a >50degree bending of the membrane is already required for just two IP3Rs, then a whole cluster interacting this way, involving progressive bending with addition of each new IP3R, would be impossible, even in ER membranes with increased curvature. As there is already a lot of information in this manuscript, I would tone down the description of this state and just mention its existence in the dataset.

Cosmetic comments:

- Figure 1 nicely shows the 4 different features used to describe the state, using 4 different colors. But in the inset a bunch of other colors show up, without description. E.g. what domain do the yellow residues in the JD Calcium binding site correspond to? Similar for the colors in the CD Calcium site (cyan residues). Line 96 also mentions the CLD, referring to panels C and D in this figure. Please make sure the CLD is indicated here. These seem to be defined in Fig 3A, so it would make more sense to put the domain description up front in fig 1 for clarity. The descriptor 'central domain' is used frequently in the text, but not shown on this figure. Make sure to clearly define which portions are part of the CD. This is particularly useful for readers not familiar with IP3Rs.

- Some movies are confusing in regards to where they're referenced in the manuscript. E.g. line 92 mentions movie M7 after mentioning that the S1-S4 density is weaker, but it's hard to assess this from that movie, and it's not clear what's different from Movie M2 mentioned just before on line 87, i.e. I don't see any differences here in the S1-S4 density from simply running the movies. The movies mostly show 3D variability analysis, which isn't mentioned in the manuscript until later. So it may be better to only reference the movies once this is discussed.

- Figures and Movies are not numbered according to how they appear in the manuscript

- Several figures have fonts that are way too small. E.g. S14 panel E just to name one.

- line 133 mentions density for IP3 in the asymmetric subclasses. Please show these densities in a figure to prove that they're indeed at full occupancy.

- calcium oscillations: line 259 mentions that the E1125Q mutation cuts the inter-spike interval approximately in half. It doesn't look like that from Fig S12F. Just include the exact interval times for WT and mutant, or otherwise mention the exact % decrease. Also, there are still oscillations, implying that there is still inhibition of IP3R activity despite removing the inhibitory binding site. This could be due to local depletion of ER calcium. As luminal calcium has been proposed to affect IP3Rs, the authors could discuss this option.

Reviewer #1 (Remarks to the Author):

In the present manuscript, authors have done cryo-EM structural and functional characterization of type 3 IP3R (hIP3R3) at varying calcium concentrations spanning the physiological range. Through this work, authors have deduced conformational landscape that enables IP3Rs to pivot from activation to inhibition to generate Ca²⁺ oscillations thereby providing structural basis for IP3R-generated Ca²⁺ oscillations. Briefly, 5 major structures for resting, preactivated, preactivated+calcium, activated and inhibited states of hIP3R3 have been identified. The 5 structures have been analyzed for their ligand binding (IP3/ATP/Calcium/Zinc) profiles and their functional and mechanistic implications in generating calcium oscillations. Intermediate states along the activation to inhibition trajectory have also been identified through advanced cryo-EM image processing methods. Section describing higher order inhibited states is intriguing.

Cryo-EM structure determination and analysis is comprehensive and is supported with mutation and other biochemistry experiments.

Comments:

We sincerely thank the reviewer for their comments and improvements to the manuscript. All references cited in our response are listed at the bottom.

1. At several places in the text, figure for the activated state (Figure 1E) is cited for the inhibited/closed state. Figure 1F corresponds to closed state. e.g.,

Lines 118-119: *"In the fifth major state and the pore is closed (Figure 1E; Tables S1 and S2)"*

Lines 194-196: *"We observed two distinct inhibited states –higher-order assemblies (Figures 1E, S4 and S2)."*

Thank you for identifying these errors. Line 119 has been modified to list the correct figure (Figure 1F).

Figure 7 can also be cited for assembled inhibited state in lines 194-196.

Line 126 has been modified as recommended by the reviewer.

Line 196 has been modified as recommended by the reviewer.

2. Lines 163-166: Appropriate figure citations from Figure 1 can be provided.

We reference Figure 1B-F on line 172-176.

"In the preactivated+Ca²⁺ state, we observed a density peak that we assigned as a Ca²⁺ in the JD site while the CD site was unoccupied (inset in Figure 1D). The Ca²⁺-binding profile of the activated state is the same as the preactivated+Ca²⁺ state, with an

occupied JD site and an empty CD site (inset in Figure 1E). Only in the inhibited state did we observe densities corresponding to Ca^{2+} in both sites (inset in Figure 1F)."

3. At several places in the text Tables are referred as Table S1, S2 etc. While the actual Tables are labeled as Table 1, 2 etc. e.g.,

We modified all references to tables in the text to match the table naming i.e. "Table 1" instead of "Table S1".

Line 69-71: *"merged these datasets and performed image processing in aggregate (Figure S2; Table S1)."*

"Using hierarchical classification, we resolved five four-fold symmetric major states for hIP3R3 at resolutions up to 2.5 Å (Table S1 and S2)."

It is not clear if above statements are referring to Tables 1 and 2.

We modified the reference above to simply Table 2, which contains statistics for the models of the five major states.

4. **Figure S10:** Mismatch between Figure panels and their description in the figure legend. e.g., Apo vs resting state in figure panel (A) is incorrectly described as resting and preactivated states in Figure text. Rest of the descriptions also do not match the Figure panels.

We added "ligand-free (apo) and resting states" to the list of sub-figure panels in the figure legend to correct the mistake.

5. **Figure 7H** description mentions *"modeling of a hypothetical higher-order interaction of the preactivated, preactivated+ Ca^{2+} , or activated state"* but only activated state is shown in Figure 7H

We have revised figure 7H to include hypothetical models for higher-order interactions between the resting, preactivated, preactivated+ Ca^{2+} and activated state tetramers.

6. High resolution for **Table 8** can be provided to make text easily readable.

We remade all tables in a clear, high-resolution format.

7. **Lines 482-484:** *"Consistent with BTF ring disruption being a key aspect of inhibition, mutations at the interface between BTF1 and BTF2 of the neighboring protomer can diminish or eliminate carbochol-induced Ca^{2+} oscillations in cells (Figure S12X)."*

Residues mutated at the interface between BTF1 and BTF2 can be mentioned. Figure S12X refers to the effect of CTD deletion mutant. Figure for the effect of mutations at the BTF1 and BTF2 interface should be provided.

We added reference to Figure 7J-L and S12R at line 484. The mutations that we examined (Trp168Ala/Lys169Ala and Lys169Ala) are listed on lines 484-497.

8. **Lines 580-584:** Suitable references supporting this statement can be provided.

We have added 6 references that support roles for post-translational modifications, protein-protein interactions, cellular metabolites, and other forms of regulation in regulating IP3R activity. We were unable to identify an appropriate reference to support a role for "membrane lipid content" in regulating IP3R activity and thus modified this to "cellular metabolites" for which there are numerous examples. We modified "can" to "may" to emphasize that the statement must be validated experimentally. The revised sentence on line 586-590 is now:

"This energetic landscape presents a highly-tunable system where post-translational modifications, protein-protein interactions, cellular metabolites, and other forms of regulation may tune the balance of states to modulate activation or alter the frequency and amplitude of Ca²⁺ waves without disturbing the principal biphasic Ca²⁺ dependence of the channel 19,21,71,55,72,73."

9. Like Figure 3Q, a figure describing CD calcium binding site can be provided and cited for Lines 168-170.

We added Figure 3R and 3S depicting the CD Ca²⁺ binding site in expanded (resting, preactivated, preactivated+Ca²⁺, and activated) and contracted (inhibited) conformations. We now reference Figure 3P-S in the text where appropriate. We added text to the figure legend: "(R-S) Superpositions of the CD Ca²⁺ binding site in the (R) Ca²⁺-free states and the (S) Ca²⁺-bound state."

10. **Table 4** description mentions "*Validation of cryo-EM density presented in Figure 3 and Figure S13*"

However, Figure 3 has no cryo-EM density described in it and Figure S13 cryo-EM density validation is not presented in Table 4. In my opinion Table 4 describes validation of cryo-EM density mentioned in Figure 4

We modified the Table 4 description to accurately describe the data. We then checked the text to ensure that the table is being correctly referenced throughout.

11. **Table 5** description mentions "*Validation of cryo-EM density presented in Figure 4 and Figure S15*"

Are the authors intending to refer to Figure 6 instead of Figure 4. Figure 4 density does not match with the description provided in Table 5.

Corrected.

12. **Table 6** description mentions “*Validation of cryo-EM density presented in Figure S17*”

Table 6 instead describes validation for density presented in Figure S13

Corrected.

13. Tables 4-6 are misleading and should be reworked.

Corrected.

14. Validation of cryo-EM density presented in Figure S17 is not provided.

Added Table 9 with this validation information, thank you for pointing that out.

15. In accordance with the Journal policy, *proper EMDB and PDB codes* should be provided in the Data and Code availability section and referenced in the manuscript and Tables.

PDB and EMDB accession codes have been added to the Data and Code availability section, which is now:

“Cryo-EM maps and atomic coordinates have been deposited with the Electron Microscopy Data Bank and Protein Data Bank, respectively: Resting state (EMD-41323 and 8TK8), Preactivated state (EMD-41347 and 8TKD), Preactivated+Ca²⁺ state (EMD-41348 and 8TKE), Activated state (EMD-41349 and 8TKF), Inhibited state (EMD-41350 and 8TKG), Labile resting state 1 (EMD-41351 and 8TKH), Labile resting state 2 (EMD-41352 and 8TKI), Higher-order inhibited state 1 (EMD-41366 and 8TLA) and Higher-order inhibited state 2 (EMD-41365 and 8TL9). Cryo-EM maps have been deposited with the Electron Microscopy: Data Bank Resting-to-Preactivated transition states (EMD-41324, EMD-41325, EMD-41326), ARM2 retractions states (EMD-41327, EMD-41328, EMD-41329, EMD-41330, EMD-41331, EMD-41332), Wedge loop progression states (EMD-41333, EMD-41334, EMD-41335, EMD-41336, EMD-41337, EMD-41338), ~C2 Preactivated TMD Transition (EMD-41339), ~C4 Preactivated TMD Transition (EMD-41340), Activated CTD states (EMD-41341, EMD-41342, EMD-41343), ~C2 Resting TMD Transition (EMD-41344), and ~C4 Resting TMD Transition (EMD-41345). Custom MATLAB scripts used for fluorescence based Ca²⁺ imaging data analysis are available at <https://github.com/vinay-sapuru/hIP3R3.git>. Summary data is available with the manuscript. Plasmids are available upon request.”

16. **Lines 321-327:** Like other places in the text, terms “*rotation*” and “*tilt*” can be quantified here.

The reconstructions presented in Figure 4B-G and described in lines 321-327 were derived from a 3DVA centered on ARM2 of a single protomer from the particles that comprise the resting-to-preactivated transitions. These reconstructions delineate a

continuum from the extended ARM2 conformation of the resting state to the retracted ARM2 conformation of the preactivated state. While we can qualitatively describe the conformational changes that we observe in the density maps, quantifying these changes requires that we build accurate atomic models.

In response to the reviewer's comment, we attempted to build atomic models into these low-resolution reconstructions by rigid-body fitting the channel into the maps as domains. However, when we closely inspected these models, we noticed that there were regions of the model that poorly fit the density. This was especially true in the more dynamic regions of the reconstruction such as ARM2, which may not move entirely as a rigid body. Due to these limitations, we cannot accurately quantify these movements and would prefer to describe the changes qualitatively. Moreover, as these conformations represent snapshots along a continuum, it is not clear if the magnitude of the frame-to-frame movements are meaningful or if they simply represent the number of bins along the continuum in which we have forced the particles.

17. Figures 4 and 7: Resolution for the cryo-EM density maps presented in these figures can be mentioned. Mention if low pass filtered to enable comparison/description.

In Figure 4, the maps are unsharpened and low-pass filtered to 5 Å, which is now noted in the figure legend. We also reference Table 4 for refinement statistics and filtering in the figure legend.

Figure 7 has been substantially revised to better show the octameric assembly. The maps are all now shown at their corresponding resolutions, which are noted in Table 2.

18. Figures 7G and S4F: Distinct hIP3R3 domains surrounding Loops 1-4 in higher order clusters can be labeled/colored to enable clear understanding of their positioning.

We have substantially revised Figure 7. In the revised version, we now show the octameric assembly in two views in panels D and F and the ordered loops in an inset in panel G. The loops in the right tetramer are depicted as sticks to provide contrast with the rest of the protein in cartoon.

19. Lines 777-779: *“Inspection of the resulting composite maps showed that they were free of model-based overfitting, for example density for ions, ligands, and **lipids remain intact** despite being removed from the input model.”*

What is the nature, binding modes, presence, or absence states of lipids in the five major states of hIP3R3 reported namely, resting, preactivated, preactivated+calcium, activated and inhibited. Mention if any correlation between lipid binding and calcium concentrations/titrations in gating mechanism across these five states?

Similar to cryo-EM reconstructions of other 6TM ion channels, we observe numerous non-protein densities that decorate the exposed hydrophobic surfaces of the TMD in the higher resolution resting, activated and inhibited states. Although less well resolved than

the nearby protein atoms, we were able to model 6 lipids in the resting and inhibited states and 5 lipids in the activated state. The lipids in the resting and inhibited states occupied overlapping positions and adopted similar conformations, consistent with their TMDs adopting nearly identical conformations. In contrast, several of the lipids in the activated state appear to be distinct from those in the resting and inhibited states. However, it is unclear if these changes are simply due to the altered shape of the protein in the activated state. Moreover, as we know of no reports of lipids directly regulating the gating of IP3Rs, the functional implications of any bound lipids are unclear.

20. **Lines 591-594:** *“Electrophysiological analyses have demonstrated that **inhibition of IP3R3 is highly cooperative, while activation is not** 19,79. These observations are consistent with the existence of **multiple asymmetric states along the trajectory to activation and the complete absence of states along the trajectory to inhibition.**”*
Lines 602-604: *“The presence of multiple transition states resolved in our analysis of **hIP3R3 thus contrasts with two prior structural titrations of the Slo2.2 and GIRK K⁺ ion channels where intermediate states were noticeably absent and the transitions from open to closed were highly cooperative processes** 46,83”*

The above two statements do not bring out the contrast as mentioned in Lines 602-604 and imply that lack of multiple states makes inhibition a highly cooperative process across hIP3R3, Slo2.2, and GIRK.

We modified these lines to more precisely state our intended conclusion. The revised text on line 608-613 is now:

“Structural titrations have been previously performed using two ligand-gated ion channels, Slo2.2 and GIRK 47,84. Intermediate states that were noticeably absent from these analyses, which led to the conclusion that the gating processes were highly cooperative 47,84. As electrophysiological analyses of Slo2.2 and GIRK also demonstrate this cooperativity, the correspondence between the structural and functional titrations of Slo2.2, GIRK, and IP-3R ion channels indicates that structural titrations can provide mechanistic insights into the processes that underlie protein function.”

21. **Lines 180-182:** *“Single-particle cryo-EM analysis of vitrified samples represents a near equilibrium assessment of their conformational landscape, allowing one to infer relative conformational free energy from the number of particles that populate specific structural classes.”*

How good is using particle counts as proxy for free energy at estimating conformational free energy landscapes compared to quantitative estimation methods?

We know of no reported analyses that have directly compared conformational free energy landscapes from particle counts with quantitative estimation methods and so it is unclear how well these two approaches compare. However, previously published structural titrations of Slo2.2 and GIRK found a high degree of correspondence between

the structural and functional titrations (Hite and MacKinnon 2017; Niu et al. 2020). For a structural titration, the relationship between particle counts and free energy relies on the assumption that the population of imaged particles faithfully samples the probability distribution adopted by the protein at a particular condition by capturing thousands of independent snapshots of the protein of interest without bias. We recognize that there are several factors that may bias the probability distribution and lead to this assumption being false.

First, vitrification of a 1 μm thin-layer specimen occurs on the order of 10^{-4} s and at a rate of 10^6 $^{\circ}\text{C}/\text{s}$, a short timescale for macromolecular rearrangements but long for diffusion of individual atoms and small molecules (Dubochet et al. 1988; Adcock and McCammon 2006). For example, a water molecule in pure water at 37 $^{\circ}\text{C}$ would have moved $\sim 13,000$ \AA in this timespan (the average displacement with self-diffusion coefficient (D) $\sim 3 \times 10^{-9}$ m^2s^{-1} calculated by $X^2=6D\Delta t$) (Holz, R. Heil, and Sacco 2000). The energy of thermal agitation at 37 $^{\circ}\text{C}$ is approximately $RT=2.5$ kJ/mol where $R=8.31$ $\text{J}/\text{mol}\cdot\text{K}^{-1}$ is the universal gas constant and $T=300.15$ K , but this drops to $RT=0.6$ kJ/mol at liquid nitrogen temperatures ($T=77$ K), below that of the ~ 1.5 kJ/mol van der Waals forces that contribute to protein structure. In this manner, large-scale conformational states are highly-preserved in vitrified specimens, but waters, ions, amino acid side chains, and even lipid acyl chains have likely adjusted to lower-temperature states driven more significantly by enthalpy. This explains why, for example, lipid acyl chains often appear well-ordered on the periphery of proteins in cryo-EM data. While we take care to image ice areas that are as thin as possible without excluding particles to minimize the ice thickness and thus the freezing rate, freezing is a necessary step of the vitrification process and one that may shift our system out of thermodynamic equilibrium.

Second, the volume of the sample changes during the vitrification as more than 99% of the buffer is wicked away from the sample by the blotting paper, which can shift the system out of thermodynamic equilibrium. The small volumes that persist after blotting are also very susceptible to dehydration, which would change the concentration of components in the buffer and also shift the system out of thermodynamic equilibrium. To minimize these effects, we incubate our samples for 30 seconds on the grid in the presence of 100% humidity to allow the system the opportunity to establish an equilibrium.

The third and perhaps greatest factor that can limit the relationship between particle counts and free energy is the particle selection itself. We did not use any 2D classification steps to cull particles as these can exclude low abundance views or conformations, thereby biasing the remaining population. The population can also be biased by artificially binning two or more distinct structural states together during 3D classification. We therefore repeated the entire classification process three times with the combined data to allow us to thoroughly investigate the conformational space before drawing our final conclusions. Each time, when a new major conformation was observed, we added it as a reference for decoy classification for the next pass through the combined datasets. Altogether, our approach enabled us to identify several particle

populations of less than 10,000 of the combined 1,700,000 initial particles that depict discrete conformational states (Figure S2) and thus we propose that our classification approach does not bias our ability to describe the underlying probability distribution of the channels.

22. **Table 3.** In the validation of higher order inhibited assembly, model composition mentions zero ATP/IP3 ligands in the higher order states. **Are these ligands not observed or not modeled into the maps?** It should be clarified in the text or table. Also, can be clarified is the reason to build only a fraction of hIP3R3 (2 chains, 11,906 atoms) in the maps for higher order inhibited states vs the isolated inhibited states (4 chains, 72,296 atoms).

The model of the higher order inhibited states was initially built in a map generated from a focused refinement that only included the interfaces of the two interacting protomers. We have now generated a model of an octameric assembly from reconstructions of two adjacent tetramers. In the combined map, densities corresponding to bound ATP molecules and occupied JD Ca²⁺ binding sites are present in all 8 protomers. In contrast, we did not observe any densities that could be attributed to a Ca²⁺ ion in the CD site. While it is possible that any ions occupying the CD Ca²⁺ site would be too disordered to be resolved, clear densities for side chains are apparent for many nearby residues suggesting that we would observe some density in the site if it were fully occupied. Because of the lower resolution of the map near the IP3-binding site, the focus refined maps were not interpretable. However, with a 7 Å low-pass filtered map, we could observe densities for IP3 molecules in each of the 8 protomers of the octomer. Thus, we modeled the assembled state with occupied ATP, IP3 and JD Ca²⁺ binding sites and an unoccupied CD Ca²⁺-binding site.

23. **Lines 203-206:** *“Although the **structures of the tetramers in the higher-order assemblies are indistinguishable from the individual inhibited tetramers**, their divergent Ca²⁺-dependence suggests that they are distinct states and that formation of higher-order assemblies may represent an alternative mechanism for achieving an inhibited state, as we will discuss later.”*

Lines 503-507: *“Although the **tetramers in these assemblies are globally quite similar to the isolated inhibited channels** and densities can be observed in both Ca²⁺-binding sites, they display an alternative Ca²⁺ dependence, suggesting that **channels in higher-order assemblies** may be functionally and **structurally distinct from isolated inhibited channels**.”*

With a complete model of an octameric assembly, we can now directly compare the two conformations of the isolated and assembled inhibited states. The RMSD between the two models is 1.2 Å with most of the differences resulting from rotations of the cytosolic domains to allow two protomers to be aligned to form the inter-tetramer interactions. We can now clarify our statements.

We have revised the text to:

Line 126-131 “In the fifth major state, the BTF ring is disrupted, ARM2 is retracted, the JD ring is intact, and the pore is closed (Figure 1F and S9; Table 2). A minor population of particles sharing these features was also identified in which the channels were organized into higher-order assemblies containing two or more tetrameric channels (Figure 7 and S10; Tables 3, 10, and 11). Notably, the interactions that mediate the assemblies are the only distinguishing feature between these two states. Otherwise, the channels adopt similar conformations..”

Line 523-527 “Outside of the interfaces and BTF1, which was too poorly ordered to model, the tetramers in the assemblies are similar to the isolated inhibited state with an all atom RMSD of 1.2 Å. The largest differences are in the flexible cytosolic domains, which rotate slightly compared to their positions in the isolated inhibited state to enable both the “+” and “-” interfaces to interact with the neighboring tetramer.”

If the complete models are not built into the higher order assemblies (comment 22), are the structural comparisons made only on density maps to state “*structures of the tetramers in the higher-order assemblies are indistinguishable from the individual inhibited tetramers*”?

The comparisons are now made using a model of the octamer.

If the structures of *tetramers in higher order assemblies are indistinguishable from the isolated inhibited tetramers* as stated in lines 203-206, what *distinct structural features of tetramers* between higher order and isolated states are being mentioned in lines 503-507?

We have now revised this section to be clearer. Because we have yet to describe the loops that are uniquely ordered in the higher order assemblies at this point in the manuscript, we have revised the sentence on line 213-216 to:
“Although the structures of the tetramers in the higher-order assemblies are very similar to the isolated inhibited tetramers, their divergent Ca^{2+} -dependence suggests that they are distinct states and that formation of higher-order assemblies may represent an alternative mechanism for achieving an inhibited state, as we will discuss later.”

24. **Lines 144-145:** “*IP3 is coordinated by the same residues in both binding modes.*”

It is interesting to see that despite ARM1 tilting towards IP3 to contact ARM1-BTF2 interface, IP3 coordinating residues remains same.

We agree! We added "Notably" in front of this sentence to improve the flow of the text.

25. **Lines 55-59:** “*We therefore sought to establish high-resolution **thermodynamic models of IP3R activation and inhibition** using single-particle cryo-EM. By collecting images of human type 3 IP3R (hIP3R3) vitrified in a broad range of Ca^{2+} concentrations and treating particle abundance as a proxy for the relative free energy of each state, we evaluate how **Ca^{2+} biases the conformational landscape of IP3Rs.***”

These results establish the structural basis for IP3R-generated Ca²⁺ oscillations.

The above statements generalize the results of this manuscript to IP3 receptors by conducting structure-based calcium titrations on type hIP3R3 receptors. Would type IP3R1 and IP3R2 channels be expected to exhibit similar conformational landscape with five major states with similar calcium binding profiles as described here for hIP3R3 receptors?

Previous analyses of type 1, type 2 and type 3 IP3Rs have reported biphasic relationships between Ca²⁺ and channel open probability. As the binding sites for IP3, Ca²⁺ and ATP are conserved between all three types, we predict that all three would display a similar conformational landscape. However, as the affinities for IP3, Ca²⁺ and ATP vary among the three types, we predict that the equilibrium parameters of the landscape would also vary.

26. Lines 171-173: *“Taken together with our previous analyses of hIP3R3 in saturating Ca²⁺ 29, these data are consistent with the **JD and CD sites being the primary Ca²⁺ binding sites in IP3Rs.**”*

Following study identifies 5 calcium binding sites (Figure 3) in type 1 IP3R (IP3R1) channels at inhibiting calcium concentrations.

Fan, G., Baker, M.R., Terry, L.E. et al. Conformational motions and ligand-binding underlying gating and regulation in IP3R channel. Nat Commun 13, 6942 (2022).

It would be interesting to see if the residues defining other three calcium binding sites (besides JD and CD) are conserved across IP3Rs. Structural comparison can shed light on differential calcium binding profile of inhibited states of IP3R3 (2 calcium binding sites) and IP3R1 (5 calcium binding sites)

To examine the proposed Ca²⁺ binding sites described in Fan et al, 2022, we downloaded 8EAQ from the Protein Data Bank. While the authors report 5 sites in the manuscript, 6 were modeled in the structure (labeled as atoms 5102-5107). Due to the discrepancy with the manuscript, we compare our 2.5 Å inhibited state structure using the atom assignments described in 8EAQ.

Ca-5102 is located in the pore of the channel. We modeled a water molecule into a density peak ~2 Å away from this position. We modeled this density as a water based on its interactions with the side chains of His2468 (2.7 Å) and Asn2472 (3.0 Å).

Ca-5103 appears to be a symmetry mate of Ca-5102.

Ca-5104 is also located in the pore, near the position of a density that we assigned as an ordered water molecule (~1 Å away). We assigned this density as water molecules due interactions with the backbone of Leu2509 (3.5 Å) and the side chain of Asn2510 (2.8 Å).

Ca-5105 is located at the interface between BTF1 and BTF2. While the structure appears similar to that of IP3R1 and the residues near the site are conserved, we did not resolve any non-protein densities in this region. However, non-protein densities

would be hard to identify due to the lower resolution of these domains compared to the pore (3.4 Å for this region in the locally refined reconstruction). For comparison, we also examined this site in the resting state structure, finding a density into which we modeled an order water ~2 Å away. This density was assigned as a water based on its interaction with the side chain of Arg53 (3.1 Å).

Ca-5106 is the JD site in our structures.

Ca-5107 is at the interface between BTF1 and BTF2 of adjacent subunits. We again compared with the resting state due to the limited resolution of the resting state in this region of the map and found an density that we assigned as an ordered water ~2 Å from Ca-5107 based on interactions with backbone of Val179 (3.1 Å) and the side chain of Asp181 (2.1 Å).

Based on the strong correspondence between Ca-5105 and Ca-5107 with ordered water molecules in the resting state, we also examined Ca-5102 and Ca-5104 in the resting state. We observe evidence of weak non-protein densities near Ca-5102, but they were too weak to assign as they may result from noise.

An ordered water molecule is also located near Ca-5104 (~1 Å away) where it is coordinated by the side chains of Asn2472 from two adjacent subunits (3.2 and 3.6 Å).

Altogether, we observed non-protein densities near to each of the Ca²⁺ binding sites proposed in Fan et al, 2022. However, only at the JD site did the structure support assigning these densities as Ca²⁺ ions. Notably, assigning non-protein densities in cryo-EM density maps is a challenging process as no specific tools have been developed such as anomalous scattering. We are therefore conservative in our assignments and have to rely on local geometry in our assignments, which can be inaccurate at the moderate resolutions of our reconstructions (worse than 2.5 Å). Future studies with improved resolutions or approaches may clarify whether these sites correspond to bound ions.

27. Are the second and third major states namely preactivated and preactivated+calcium states observed for the first time in this manuscript or similar structures are available in the literature? Resting, activated and inhibited states of hIP3R3 reported here are mentioned to be observed in previous studies on IP3Rs.

The preactivated state has been previously observed by our group and others.

We added the following sentence on line 127 to clarify:

"The preactivated state, with its intact BTF ring and retracted ARM2 domain and closed pore, is similar to previously reported structures of IP3Rs in the presence of IP3 33,37 (PDB: 6DQV, 7T3P, 7T3Q, 7T3R), but the preactivated+Ca²⁺ state not been previously described."

Reviewer #2 (Remarks to the Author):

This manuscript describes the conformational landscape of the IP3 Receptor, a well-known calcium release channel that resides in the ER membrane. The authors used cryo-EM to perform titrations of the IP3R across a range of free calcium concentrations,

carefully analyzing the distribution of particles in various distinct states. The authors nicely show that various states already exist at different concentrations, and that increasing levels of calcium change the relative distributions. They also provide very detailed descriptions of the structural transitions and show the importance of the different calcium binding sites using calcium imaging experiments.

Although many high-resolution cryo-EM studies of the IP3R have been described before, this is by far the most detailed and systematic study of IP3R structure and conformation, and even includes the discussion of asymmetric states. This is of high interest in the field, although I think that, as presented, the manuscript is also very dense in several sections, which may deter experts in the IP3R field who are not structural biologist themselves.

The study deserves a home in Nature Communications, but I have a series of suggestions for improvement.

Comments:

We sincerely thank the reviewer for their comments and improvements to the manuscript. All references cited in our response to the review are listed at the bottom.

* The authors went to great care to avoid contaminating calcium from the blotting paper. However, it not clear what exactly has been done to experimentally verify the predicted free calcium concentration in the buffers. There is a short section “fura-2 calibration” in the methods, and Fig S1 shows fluorescence, but no experimentally determined concentrations are presented. The authors should list the measured free calcium concentrations in their buffers and include this in the Results. This could be checked with a calcium-sensing electrode, or with a calcium indicator, but using reference standards to obtain true calibration curves. If some of the calcium concentrations fall outside of the testable range, this should also be mentioned explicitly.

As suggested by the reviewer, we used a Fura-2 calibration kit (Invitrogen) to calculate a standard curve for Fura-2 fluorescence from 17 nM to 1.35 μ M. The 340/380 nm emission ratio values for the nominally 1, 10 and 100 nM Ca²⁺ chelator cocktail samples were interpolated on the calibration curve to estimate free Ca²⁺ concentrations of 39.4, 49.0 and 155.1 nM, respectively. The 340/380 nm emission ratio values for the nominally 1 and 10 μ M Ca²⁺ chelator cocktail samples were outside of the standard curve and therefore had to be extrapolated from the calibration curve to estimate free Ca²⁺ concentrations of 2.77 and 7.58 μ M, respectively.

Free Ca²⁺ concentrations estimated from WEBMAXC and those estimated using Fura-2 are now tabulated and presented in revised Figure S1. Notably, it is unlikely that either of these estimates correspond exactly to the final Ca²⁺ concentrations on the grid due to changes in temperature, buffer dehydration and pressure that occur during vitrification. Our goal was to establish a range of Ca²⁺ concentrations over which we

could observe a biphasic relationship between Ca^{2+} concentration and channel open probability, as noted in the next point.

Note: Although there is biophysical interest in investigating conformations at 1nM free calcium, there is little physiological relevance as this is much lower than the resting calcium concentration in most cells. I recommend this rationale (biophysical purposes) to be mentioned up front. The authors could also point out resting calcium concentrations that IP₃R₃ would encounter in a native context, and indicate what this would mean for the expected conformational landscape in those conditions.

We agree with the reviewer that 1 nM Ca^{2+} is outside of the physiological range of most cells and that we have selected this range to sample a range where electrophysiological analysis predicts that hIP₃R₃ would display a biphasic relationship with Ca^{2+} . We have revised the text to convey this point as follows:

From Line 66, we removed "physiological". We have revised the subsequent sentence to:

"Our cryo-EM conditions correspond to a range where electrophysiological analyses would predict that hIP₃R₃ displays a biphasic relationship between Ca^{2+} concentration and channel open probability."

To emphasize that the basal Ca^{2+} concentration in cells is sufficiently high to bind to the JD site in the preactivated state, we revised line 564-567 in the discussion to:

"The increased affinity of the preactivated state for Ca^{2+} would promote binding of basal cytosolic Ca^{2+} at the JD site, unlocking the JD ring and favoring the transition through the ensemble of high-energy intermediate states along the trajectory to the fully-open activated state."

*The description of the calcium occupancy seems to be all or none, i.e. it was observed, or it was not observed. One can imagine that binding or not binding to a particular site correlates with the conformational state, but one would also expect asymmetry in at least some datasets that are not saturating, i.e. not all 4 equivalent binding sites in the tetramer are occupied simultaneously. Alternatively, when the degree of cooperativity in the four equivalent sites is extremely high, partial binding may never be observed in any individual class. The authors have discussed this, mentioning that high cooperativity is known to exist for inhibition, but not for activation by calcium. Thus, one would expect some classes with less than four, but more than zero calcium ions bound at the activating binding site in some of the datasets. This could manifest as either lower apparent occupancy (when C4 symmetry is applied), or, potentially, as subtle asymmetric features in the subunits (for which data would have to be analyzed in C1).

I recommend the authors to describe the symmetry/asymmetry a bit better and further describe the possibility for partial occupancy of calcium. First, what criterion did the authors use to decide whether or not there is symmetry? Most classes are described as symmetric, and some classes with clear asymmetry were observed. Was any quantitation of asymmetry utilized prior to deciding whether it was C4?

For sites showing calcium density, what was the threshold level needed to observe the calcium? And what is the density relative to the average density (overall or local in the region around the calcium)? Using this, are there signs of lowered calcium occupancy in some classes? If symmetry is not forced in the symmetric classes, are there indications for different calcium occupancies in the equivalent sites, possibly with additional subtle asymmetric features? This analysis may be outside of the limits of what could be dissected from the datasets, but I urge the authors to attempt this, or mention the results if this was already done with negative results, mentioning the limitations.

Note: Descriptions of the density thresholds used are also missing in most figures.

We thank the reviewer for highlighting the symmetry of the ligand-binding sites as it greatly interested us as well. As we had seen asymmetric states in our previous study, we avoided symmetrization whenever possible during image processing and evaluated the effect of symmetrization when it was employed. Here, we summarize the steps during processing with regards to symmetrization.

First, we did not employ symmetry during any of the 3D classification steps, nor did we use 2D classification which can result in low abundance views or conformations being discarded. In the revised version, we now note in the results section and the methods section that all heterogeneous classification steps were performed without imposing symmetry. For example, we now state on line 71-78:

“Using hierarchical classification without imposing symmetry, we resolved five four-fold symmetric major states for hIP3R3 at resolutions up to 2.5 Å (Table 2). By relaxing our assumption of C4 symmetry and computing latent representations of the conformational heterogeneity present in the remaining classes using 3D variability analysis (3DVA) 36, we were also able to reconstruct discrete low-abundance intermediates, including several that are asymmetric. Following classification, we improved the interpretability of the reconstructions by performing symmetry expansion and local refinements that were subsequently merged into composite reconstructions.”

Second, we calculated reconstructions for each state with and without imposing symmetry. We did not perform a quantitative assessment of asymmetry. Instead, we assessed if the imposed symmetry was appropriate by comparing maps with and without symmetry imposed to determine if the features of the maps were improved or degraded by symmetrization.

Third, we further assessed the effect of symmetrization on each state by performing symmetry expansion, where each of the four protomers is treated as a unique particle for focused classification using masks encompassing different regions of the channel. For the JD Ca²⁺-binding site, we used a mask including ARM3 and the JD. Several distinct states could be observed in the results for classification of the more dynamic regions of the channel, such as ARM2. However, we did not observe any appreciable differences in the densities for the Ca²⁺ ions. In addition to allowing us to perform focused classifications, the local refinements of the symmetry-expanded particles also greatly improved the resolution and interpretability of the reconstructions. The improved maps were critical to evaluating the Ca²⁺ binding sites, as ions are poorly resolved in

maps at resolutions worse than 4 Å such as the C1 preactivated+Ca²⁺ global consensus map. Thus, while it is possible that we imaged particles with substoichiometric ion binding profiles, we were not able to identify any during image processing.

Characterizing the occupancy of a ligand in EM maps is quite challenging, especially for the moderate resolutions of our reconstructions (>2.5 Å). As suggested by the reviewer, we compared the threshold at which the ions could be observed in the maps with the surrounding protein atoms and found that the densities corresponding to the bound Ca²⁺ ions are slightly weaker than nearby amino acid side chains in the composite maps for the preactivated+Ca²⁺, activated and inhibited state. In the preactivated+Ca²⁺ state, the JD site density can be observed at 5.5 sigma, while densities for nearby sidechain atoms are observable at 7-8 sigma, with ARM3 having stronger densities than the JD. In the activated state, the JD site density can be observed at 5.5 sigma, while densities for nearby sidechain atoms are observable at 7-8 sigma, with ARM3 having stronger densities than the JD. Notably for both of these maps, there are backbone atoms in nearby flexible regions of the channel, such as residues 1718-1720, that cannot be observed at 5.5 sigma. For the inhibited state, the JD site density can be observed at 11 sigma, while densities for nearby sidechain atoms are observable at 12-13 sigma. The CD side in the inhibited state can be observed at 6 sigma, while nearby sidechain atoms in the CLD can be observed at 10 sigma and sidechain atoms in ARM2 can be observed at 4 sigma. From this analysis it is unclear if the occupancy for the Ca²⁺-binding sites in the preactivated+Ca²⁺ or activated states are lower.

We have revised the legend for Figure 1 to include the threshold for each inset. Note that these maps are regions of a composite map and thus threshold can only be compared very locally with other segments from the same locally-refined map. The revised figure legend is now:

Figure 1: Structural Ca²⁺ titration of hIP3R3. (A) Schematic for cryo-EM Ca²⁺ titration of hIP3R3. (B-F) C4-symmetrized composite cryo-EM density maps viewed from the cytosol (left) and the side (right) with structural heuristics (top-right corner) and ligand binding status (bottom insets for IP3, CD Ca²⁺, JD Ca²⁺, and ATP) for the (B) resting, (C) preactivated, (D) preactivated+Ca²⁺, (E) activated, and (F) inhibited states. Insets in B-F are colored by domain: BTF1 (purple), BTF2 (blue), ARM1 (light blue), CLD (cyan), ARM2 (green), ARM3 (yellow), JD (orange), and TMD (red). Insets in (B) are contoured at 4, 8, 9 and 4 thresholds for the IP3, CD Ca²⁺, JD Ca²⁺, and ATP sites, respectively. Insets in (C-E) are contoured at 5, 7, 9 and 4 thresholds for the IP3, CD Ca²⁺, JD Ca²⁺, and ATP sites, respectively. Insets in (F) are contoured at 1, 9, 15 and 6 thresholds for the IP3, CD Ca²⁺, JD Ca²⁺, and ATP sites, respectively.

* readability: several sections are very dense, especially the descriptions on pages 13-16. This may deter many non-structural biologists in the IP3R field. Details matter, but the authors may want to provide more overall descriptive summaries in each section of the results, followed by the details (or move some details to supplementary and/or figure legends).

We thank the review for their suggestion. We have substantially revised the section “Activation of hIP3R3 by IP3, Ca²⁺ and ATP” in an effort to improve readability for a general audience.

*The ‘assembled inhibited state’ is bizarre, and it is not clear if there is physiological relevance. In the ER membrane, there are 2D constraints imposed by the membrane. Figure 7E and 7F show that there would need to be an extreme curvature of the membrane that allows this, and this result may thus be an artefact from the preparation, where no membrane is present. Clustering of IP3Rs in ER membranes has been observed, but if a >50degree bending of the membrane is already required for just two IP3Rs, then a whole cluster interacting this way, involving progressive bending with addition of each new IP3R, would be impossible, even in ER membranes with increased curvature. As there is already a lot of information in this manuscript, I would tone down the description of this state and just mention its existence in the dataset.

We agree that the resolved the assembled inhibited state was an unexpected finding and that the physiological relevance of this state is currently unclear. We have revised the discussion to avoid overinterpretation. The discussion of the higher-order assemblies on line 614-621 is now:

“It has long been appreciated that IP3Rs can function in clusters of multiple channels in cells that display a high degree of synchronicity 80. These clusters arise in an IP3- and Ca²⁺-dependent fashion 81,82. While it is unclear if the higher-order assemblies of inhibited channels that we observe in our analysis correspond to the clusters that have been observed in cells, they share a common ligand binding profile. As the Ca²⁺-dependency of the inhibited particles in higher-order assemblies differs from those of isolated inhibited state particles, it will be interesting to investigate if the formation of higher-order assemblies contributes to Ca²⁺-dependent inhibition in cells.”

Cosmetic comments:

- Figure 1 nicely shows the 4 different features used to describe the state, using 4 different colors. But in the inset a bunch of other colors show up, without description. E.g. what domain do the yellow residues in the JD Calcium binding site correspond to? Similar for the colors in the CD Calcium site (cyan residues). Line 96 a Iso mentions the CLD, referring to panels C and D in this figure. Please make sure the CLD is indicated here. These seem to be defined in Fig 3A, so it would make more sense to put the domain description up front in fig 1 for clarity. The descriptor ‘central domain’ is used frequently in the text, but not shown on this figure. Make sure to clearly define which portions are part of the CD. This is particularly useful for readers not familiar with IP3Rs.

Thank you for noting that the labels were incomplete. The yellow residues correspond to ARM3 and the cyan residues correspond to the CLD. We revised the legend of Figure 1 to include the following statement:

"Insets in B-F are colored by domain: BTF1 (purple), BTF2 (blue), ARM1 (light blue), CLD (cyan), ARM2 (green), ARM3 (yellow), JD (orange), and TMD (red)."

- Some movies are confusing in regards to where they're referenced in the manuscript. E.g. line 92 mentions movie M7 after mentioning that the S1-S4 density is weaker, but it's hard to assess this from that movie, and it's not clear what's different from Movie M2 mentioned just before on line 87, i.e. I don't see any differences here in the S1-S4 density from simply running the movies. The movies mostly show 3D variability analysis, which isn't mentioned in the manuscript until later. So it may be better to only reference the movies once this is discussed.

The movies depicting the 3D variability analysis were cited in the text to acknowledge that the channel remains flexible in all of the major states. We agree that inclusion was not clear from the text in the results section. We have a section to the methods section starting on line 769 to better convey this point and instead cite the movies there: "Notably, local conformational changes can still be observed in the five major states by 3DVA at the end of the hierarchical classification due the large size and overall flexibility of the channel. Our final maps and the corresponding models correspond to the average of the particles that comprise these states."

- Figures and Movies are not numbered according to how they appear in the manuscript

We have modified all identified erroneous labeling for movies and figures.

- Several figures have fonts that are way too small. E.g. S14 panel E just to name one.

We have reorganized many of the supplementary figures to improve readability. Sequence alignments are displayed them horizontally to increase font size by ~30%.

- line 133 mentions density for IP3 in the asymmetric subclasses. Please show these densities in a figure to prove that they're indeed at full occupancy.

As mentioned above for the section on the occupancy of the Ca^{2+} binding sites, there are no tools available in single-particle cryo-EM to quantitatively measure ligand occupancy. By comparing the maps at different thresholds, we can evaluate the relative intensity of the density peaks. For IP3 in asymmetric classes, the IP3 density has a similar threshold as the amino acid side chains, suggesting that the sites are well occupied. We have added a new Figure S11 depicting the density for the three asymmetric resting-to-preactivated transition states and highlighting peaks corresponding to the bound IP3 molecules. We reference Figure S11 on line 143 when discussing the IP3-occupancy of the asymmetric intermediate states for the first time.

- calcium oscillations: line 259 mentions that the E1125Q mutation cuts the inter-spike interval approximately in half. It doesn't look like that from Fig S12F. Just include the exact interval times for WT and mutant, or otherwise mention the exact % decrease. Also, there are still oscillations, implying that there is still inhibition of IP3R activity despite removing the inhibitory binding site. This could be due to local depletion of ER

calcium. As luminal calcium has been proposed to affect IP₃Rs, the authors could discuss this option.

We modified the text to more accurately state the data, changing "approximately half" to "nearly half" at line 267. It now reads as follows:

"While the mean rising phase was similar to wild-type hIP₃R3 (Figure 3G, Table 8), the mean inter-spike interval was nearly half (59%) at 12.7 seconds (Figure S14F, Table 8), suggesting that perturbing the CD site alters gating of hIP₃R3."

The exact numbers are stated for WT and CD mutant in the text at lines 261 and 268 respectively. In Fig S12F, we show individual values and the spread in the data. To assist in comparison of these mutants, we added a reference to Table 8 that summarizes the values obtained from calcium imaging analysis for the WT hIP₃R3 as well as the various mutants analyzed.

- Adcock, Stewart A., and J. Andrew McCammon. 2006. "Molecular Dynamics: Survey of Methods for Simulating the Activity of Proteins." *Chemical Reviews* 106 (5): 1589–1615. <https://doi.org/10.1021/cr040426m>.
- Bock, Charles W., Amy Kaufman, and Jenny P. Glusker. 1994. "Coordination of Water to Magnesium Cations." *Inorganic Chemistry* 33 (3): 419–27. <https://doi.org/10.1021/ic00081a007>.
- Cabra, Vanessa, Takashi Murayama, and Montserrat Samsó. 2016. "Ultrastructural Analysis of Self-Associated RyR2s." *Biophysical Journal* 110 (12): 2651–62. <https://doi.org/10.1016/j.bpj.2016.05.013>.
- Carafoli, Ernesto, and Joachim Krebs. 2016. "Why Calcium? How Calcium Became the Best Communicator *." *Journal of Biological Chemistry* 291 (40): 20849–57. <https://doi.org/10.1074/jbc.R116.735894>.
- Dubochet, Jacques, Marc Adrian, Jiin-Ju Chang, Jean-Claude Homo, Jean Lepault, Alasdair W. McDowell, and Patrick Schultz. 1988. "Cryo-Electron Microscopy of Vitriified Specimens." *Quarterly Reviews of Biophysics* 21 (2): 129–228. <https://doi.org/10.1017/S0033583500004297>.
- Fan, Guizhen, Mariah R. Baker, Lara E. Terry, Vikas Arige, Muyuan Chen, Alexander B. Seryshev, Matthew L. Baker, Steven J. Ludtke, David I. Yule, and Irina I. Serysheva. 2022. "Conformational Motions and Ligand-Binding Underlying Gating and Regulation in IP3R Channel." *Nature Communications* 13 (1): 6942. <https://doi.org/10.1038/s41467-022-34574-1>.
- Fan, Guizhen, Mariah R. Baker, Zhao Wang, Alexander B. Seryshev, Steven J. Ludtke, Matthew L. Baker, and Irina I. Serysheva. 2018. "Cryo-EM Reveals Ligand Induced Allostery Underlying InsP3R Channel Gating." *Cell Research* 28 (12): 1158–70. <https://doi.org/10.1038/s41422-018-0108-5>.
- Hattne, Johan, Dan Shi, Calina Glynn, Chih-Te Zee, Marcus Gallagher-Jones, Michael W. Martynowycz, Jose A. Rodriguez, and Tamir Gonen. 2018. "Analysis of Global and Site-Specific Radiation Damage in Cryo-EM." *Structure* 26 (5): 759-766.e4. <https://doi.org/10.1016/j.str.2018.03.021>.
- Hite, Richard K., and Roderick MacKinnon. 2017. "Structural Titration of Slo2.2, a Na⁺-Dependent K⁺ Channel." *Cell* 168 (3): 390-399.e11. <https://doi.org/10.1016/j.cell.2016.12.030>.
- Holz, Manfred, Stefan R. Heil, and Antonio Sacco. 2000. "Temperature-Dependent Self-Diffusion Coefficients of Water and Six Selected Molecular Liquids for Calibration in Accurate 1H NMR PFG Measurements." *Physical Chemistry Chemical Physics* 2 (20): 4740–42. <https://doi.org/10.1039/B005319H>.
- Katz, Amy Kaufman, Jenny P. Glusker, Scott A. Beebe, and Charles W. Bock. 1996. "Calcium Ion Coordination: A Comparison with That of Beryllium, Magnesium, and Zinc." *Journal of the American Chemical Society* 118 (24): 5752–63. <https://doi.org/10.1021/ja953943i>.
- Luzzi, Veronica, Christopher E. Sims, Joseph S. Soughayer, and Nancy L. Allbritton. 1998. "The Physiologic Concentration of Inositol 1,4,5-Trisphosphate in the Oocytes of *Xenopus Laevis* *." *Journal of Biological Chemistry* 273 (44): 28657–62. <https://doi.org/10.1074/jbc.273.44.28657>.

- Missiaen, Ludwig, Humbert De Smedt, Jan B. Parys, Ilse Sienaert, Sara Vanlingen, and Rik Casteels. 1996. "Threshold for Inositol 1,4,5-Trisphosphate Action (*)." *Journal of Biological Chemistry* 271 (21): 12287–93. <https://doi.org/10.1074/jbc.271.21.12287>.
- Niu, Yiming, Xiao Tao, Kouki K Touhara, and Roderick MacKinnon. 2020. "Cryo-EM Analysis of PIP2 Regulation in Mammalian GIRK Channels." Edited by Merritt Maduke, Kenton J Swartz, Bruce P Bean, Sudha Chakrapani, and Ryan E Hibbs. *ELife* 9 (August): e60552. <https://doi.org/10.7554/eLife.60552>.
- Paredes, R. Madelaine, Julie C. Etzler, Lora Talley Watts, and James D. Lechleiter. 2008. "Chemical Calcium Indicators." *Methods (San Diego, Calif.)* 46 (3): 143–51. <https://doi.org/10.1016/j.ymeth.2008.09.025>.
- Uto, A., H. Arai, and Y. Ogawa. 1991. "Reassessment of Fura-2 and the Ratio Method for Determination of Intracellular Ca²⁺ Concentrations." *Cell Calcium* 12 (1): 29–37. [https://doi.org/10.1016/0143-4160\(91\)90082-p](https://doi.org/10.1016/0143-4160(91)90082-p).
- Wang, J., Z. Liu, J. Frank, and P. B. Moore. 2018. "Identification of Ions in Experimental Electrostatic Potential Maps." *IUCrJ* 5 (4): 375–81. <https://doi.org/10.1107/S2052252518006292>.
- Wang, Jimin, S. Kundhavai Natchiar, Peter B. Moore, and Bruno P. Klaholz. 2021. "Identification of Mg²⁺ Ions next to Nucleotides in Cryo-EM Maps Using Electrostatic Potential Maps." *Acta Crystallographica. Section D, Structural Biology* 77 (Pt 4): 534–39. <https://doi.org/10.1107/S2059798321001893>.

REVIEWERS' COMMENTS

Reviewer #1 (Remarks to the Author):

The authors have addressed all the comments in a comprehensive manner. I strongly recommend the manuscript for publication.

Reviewer #2 (Remarks to the Author):

The authors have addressed all of my previous suggestions, and seem to have done a very thorough job of revising the manuscript, improving clarity, and adding extra descriptions and quantifications.

As noted before, this is an important study that is first in its kind in regards to a systematic investigation of the conformational landscape of a calcium release channel. I recommend publication in its current form.